# SPRINT: Scalable Semantic Policy Pre-Training via Language Instruction Relabeling

## Abstract

We propose SPRINT, a scalable offline policy pre-training approach based on natural language instructions. SPRINT pre-trains an agent's policy to execute a diverse set of semantically meaningful skills that it can leverage to learn new tasks faster. Prior work on offline pre-training required tedious manual definition of pre-training tasks or learned semantically meaningless skills via random goal-reaching. Instead, our approach SPRINT (**S**calable **P**re-training via **R**elabeling Language **INsT**ructions) leverages natural language instruction labels on offline agent experience, collected at scale (e.g., via crowd-sourcing), to define a rich set of tasks with minimal human effort. Furthermore, by using natural language to define tasks, SPRINT can use large language models to automatically expand the initial task set. As a result, we can learn an extensive collection of new skills via offline RL during pre-training by relabeling and aggregating task instructions, even across multiple trajectories. Experiments in ALFRED, a realistic household simulator, show that agents pre-trained with SPRINT learn new long-horizon household tasks substantially faster than with previous pre-training approaches.

## 1 Introduction

When humans learn a new task, *e.g.*, how to cook a new dish, we rely on a large repertoire of previously learned *skills*, like "*chopping vegetables*" or "*boiling pasta*", that make learning more efficient. Improving learning efficiency is crucial for practical deployment of artificial agents; thus, many works in reinforcement learning (RL) aim to equip agents with a similar set of skills. To autonomously acquire such skills, recent works optimize for diverse agent behaviors (Eysenbach et al., 2019; Sharma et al., 2020; Mendonca et al., 2021), imitate short action sequences (Lynch et al., 2020; Pertsch et al., 2020), or reach randomly sampled goal states (Chebotar et al., 2021) from pre-collected experience. However, such objectives may result in the agent learning skills that are not semantically plausible in practice, *e.g.*, "*placing a knife in the microwave*" or "*half-closing the microwave door*." To focus pre-training on *plausible* skills, one could instead manually curate a set of pre-training tasks for the policy, but this requires tedious reward function design and does not scale well beyond a few dozen tasks (Yu et al., 2019). Yet, defining a large set of pre-training tasks is crucial: only a policy with a wide range of skills can accelerate learning on *many* downstream tasks. How can we define a *large* set of meaningful pre-training tasks in a scalable manner?

In this paper, we propose to leverage *natural language instructions* to define a large number of semantically meaningful tasks for policy pre-training. Natural language has recently been used to allow humans to effectively interact with agents (Lynch & Sermanet, 2021) or to generate long-horizon plans (Ahn et al., 2022). In the context of defining pre-training tasks, using natural language has two important benefits: (1) language is a natural and expressive interface for humans to specify tasks (in contrast to, *e.g.*, numerical reward functions) as it is the primary way to communicate tasks in our everyday lives. Thus, even non-experts can define tasks easily via language instructions. (2) By specifying pre-training tasks via natural language, we can leverage the knowledge captured in large language models to automatically generate more tasks through instruction relabeling.

To combine both benefits we introduce SPRINT (**S**calable **P**re-training via **R**elabeling Language **INsT**ructions), a scalable pre-training approach that equips policies with a repertoire of semantically meaningful skills (see Figure 1 for an illustration). SPRINT has three core components: (1) language-conditioned offline RL, (2) LLM-based skill aggregation and (3) cross-trajectory skill

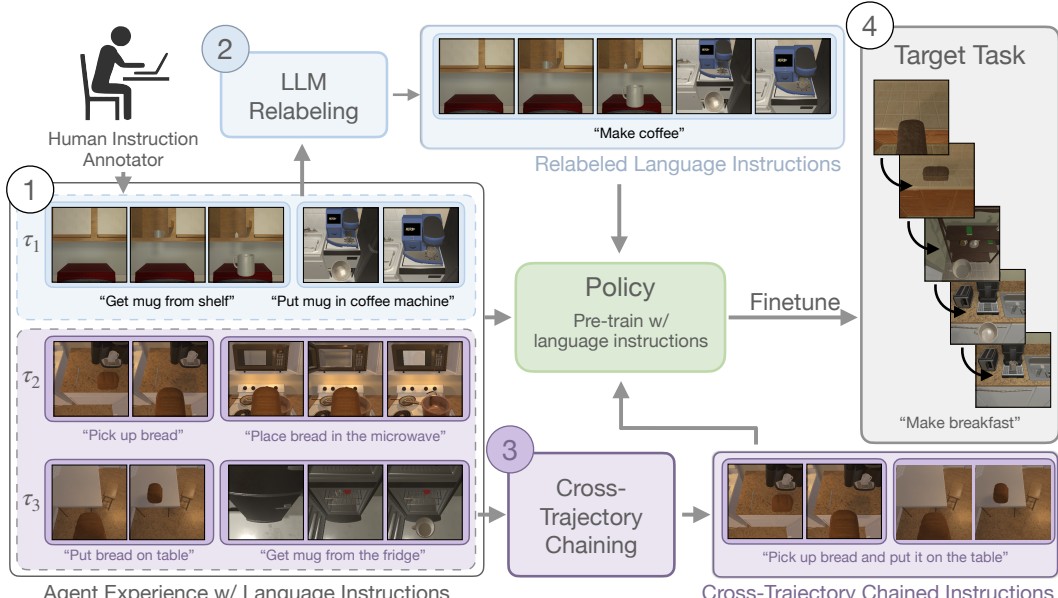

Figure 1: We propose SPRINT, a scalable approach for policy pre-training with semantic skills. We assume access to an offline dataset of agent experience with natural language instruction labels of the performed skills, *e.g.*, provided by human annotators (1). We use the instructions to pre-train a semantic skill policy via instruction-conditioned offline RL. To increase pre-training task diversity, we automatically generate new instructions via (2) language-model-based instruction relabeling, and (3) cross-trajectory skill chaining. We demonstrate that an agent pre-trained with SPRINT can leverage the diverse set of learned semantic skills to finetune efficiently on unseen target tasks (4).

chaining. SPRINT assumes access to an offline dataset of state-action trajectories, each of which performs one or more *skills*. We assume that the data has corresponding natural language instruction labels for the performed skills, such as "*place mug in coffee machine*" or "*press brew button*". Such labels can be crowd-sourced from non-expert human annotators at scale. We use these annotations as task instructions and train a policy with language-conditioned offline RL to solve them. Crucially, SPRINT uses two techniques to expand this initial task set: firstly, we use a pre-trained large language model to relabel the language instructions, thereby creating new tasks. For example, the tasks "*place mug in coffee machine*" and "*press brew button*" can be combined into a new task: "*make coffee.*" Secondly, we chain behaviors *across* multiple trajectories from the training data; starting with a skill like "*pick up bread*" from one trajectory and ending with "*place bread on table*" from another. This allows the policy to learn semantic skills completely unseen in the training data.

SPRINT trains a policy on the *combined* set of task instructions, thereby equipping the agent with a policy that can execute a wide range of semantically meaningful skills. Our experiments demonstrate that this allows for substantially more sample-efficient learning and better zero-shot execution of new downstream tasks like "*prepare breakfast*" than prior pre-training approaches.

In summary, our contributions are threefold: (1) we propose SPRINT, which leverages natural language instructions for scalable policy pre-training via instruction-conditioned offline RL, (2) we expand the set of pre-training tasks via LLM-based skill relabeling and cross-trajectory chaining, (3) we demonstrate that SPRINT enables agents to more efficiently learn long-horizon household tasks in the ALFRED simulator (Shridhar et al., 2020) than prior pre-training approaches.

## 2 RELATED WORK

**Language in RL.** There is a long-standing interest in leveraging natural language during behavior learning, *e.g.*, to structure agent's internal representations (Andreas et al., 2017b), to learn to interact with text-based games (Narasimhan et al., 2015; Küttler et al., 2020), or guide long-horizon task learning via recipe-like plans (Branavan et al., 2009; Andreas et al., 2017a). The recent progress

in training large, general-purpose language models has enabled approaches that directly generate and execute such plans solely from a high-level task description (Huang et al., 2022a; Ahn et al., 2022; Huang et al., 2022b). Others have shown how language in combination with vision inputs can be used to learn state representations (Nair et al., 2022) or reward functions (Fan et al., 2022). Finally, there is a rich body of work that explores using language as an intuitive agent interface for humans: given a free-form natural language command they aim to train language-conditioned policies that can execute the instruction in the environment (Lynch & Sermanet, 2021). In contrast, our work aims to leverage language for task-agnostic *agent pre-training*. Instead of training a policy to execute human language commands, we aim to equip an agent with a set of semantic skills that improves the efficiency of reinforcement learning on downstream tasks. In a similar vein, Colas et al. (2020) explored using language to define tasks for semantically meaningful exploration in RL, but they required a hand-defined grammar for task instruction generation, while we leverage offline data and large language models for automatic generation of new training tasks.

**Pre-training Policies for RL.**   Developing policy pre-training approaches for faster downstream learning has been investigated for many years (Ijspeert et al., 2002; Theodorou et al., 2010; Hester et al., 2018). Recent advances in offline reinforcement learning (Levine et al., 2020) enabled approaches that can pre-train agents offline and effectively finetune them on online tasks (Peng et al., 2019; Singh et al., 2020; Nair et al., 2020; Kostrikov et al., 2022). However, these approaches require the pre-training data to have target-task reward annotations and the resulting pre-trained policies are trained only to solve the target task. In contrast, meta-RL approaches pre-train on a range of tasks and allow fast adaptation to *unseen* downstream tasks (Duan et al., 2016; Finn et al., 2017; Rakelly et al., 2019; Nam et al., 2022), yet require tedious manual definition of pre-training tasks by experts, making them less scalable. To avoid manual task design, other works have explored unsupervised pre-training approaches based on behavior diversification (Achiam et al., 2018; Eysenbach et al., 2019; Sharma et al., 2019) or extraction of behavior priors from offline agent experience (Pertsch et al., 2020; Ajay et al., 2020; Singh et al., 2021). Closest to ours is the approach of Chebotar et al. (2021), which performs unsupervised pre-training by learning to reach randomly sampled states from offline agent experience. Yet, unsupervised pre-training approaches learn semantically meaningless skills which, as we demonstrate in Section 5, lead to worse transfer to downstream tasks. In contrast, we introduce a *scalable* pre-training approach based on natural language instructions that equips agents with *semantically meaningful* skills and allows for efficient transfer to unseen tasks.

## 3   PRELIMINARIES

**Offline RL.** Offline reinforcement learning (RL) methods assume access to datasets of trajectories $\tau$ composed of ordered $(s, a, s', r)$ tuples of current state, action, next state, and reward labels, from a Markov Decision Process $\mathcal{M} = (\mathcal{S}, \mathcal{A}, P, \mathcal{R}, \gamma)$. $\mathcal{S}$ and $\mathcal{A}$ are state and action spaces. $P : \mathcal{S} \times \mathcal{A} \times \mathcal{S} \to \mathbb{R}_+$ represents the transition probability distribution. $\mathcal{R} : \mathcal{S} \times \mathcal{A} \times \mathcal{S} \to \mathbb{R}$ denotes the reward function, and $\gamma$ is the discount factor. These methods learn offline, with no environment interactions, and aim to train a policy $\pi(a|s)$ that maximizes the return per episode. Temporal-difference (TD) learning methods also train a critic, $Q^\pi(s, a)$, representing the discounted future sum of rewards using policy $\pi$ after taking action $a$ from the state $s$ (Sutton & Barto, 2018).

## 4   SPRINT: SCALABLE SEMANTIC POLICY PRE-TRAINING WITH LANGUAGE INSTRUCTIONS

In this work, we propose SPRINT (**S**calable **P**re-training via **R**elabeling Language **IN**s**T**ructions), an approach for pre-training a policy to solve a wide range of semantic skills in order to enable efficient finetuning on unseen tasks. SPRINT has three core technical components which we will describe in this section: (1) offline instruction-conditioned RL with crowd-sourced language instructions, (2) language-model-based skill aggregation and (3) cross-trajectory skill chaining. But first, we will detail the data we use for pre-training and give an intuitive overview of our approach.

**Pre-training Data.**   Following prior work on agent pre-training, we assume access to a large offline dataset $\mathcal{D}$ of agent experience (Gupta et al., 2019; Lynch et al., 2020; Pertsch et al., 2020; Chebotar et al., 2021; Ebert et al., 2022; Pertsch et al., 2021). Such data can be collected at scale,

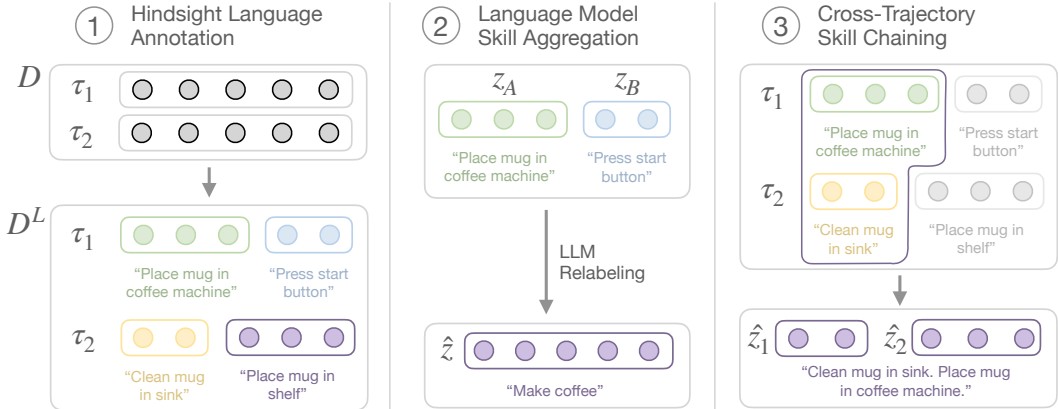

Figure 2: SPRINT overview. **Left**: Hindsight annotation of agent experience with language instructions is a scalable approach for defining pre-training tasks. We pre-train via instruction-conditioned offline RL with a sparse goal-reaching reward (Section 4.1). **Middle**: We expand the set of pre-training tasks by aggregating language instructions with an LLM and adding the relabeled trajectories back into the pre-training dataset (Section 4.2). **Right**: We perform cross-trajectory chaining of skills to enable pre-training of skills that are unseen in the offline agent experience (Section 4.3).

*e.g.*, from prior RL runs, via teleoperation, through autonomous agent exploration or any combination thereof. We further assume that the data contains natural language descriptions of the skills performed by the agent, *e.g.*, "*put a mug in the coffee machine*" or "*push the brew button.*" Such descriptions can be collected at scale by non-experts by annotating sequences from our offline dataset $\mathcal{D}$ *in hindsight*, *e.g.*, via crowd-sourcing through platforms like Amazon Mechanical Turk (Lynch & Sermanet, 2021; Shridhar et al., 2020). Given a randomly sampled sequence $\tau$ from the dataset $\mathcal{D}$, annotators can label sub-trajectories $\tau_1 = [s_0, a_0, s_1, \ldots], \tau_2 = \ldots$ with free-form language descriptions $z_1, z_2, \ldots$ of the tasks performed in the respective sub-trajectories, for example:

$$\tau_{\bar{z}} = [\tau_{z_1}, \tau_{z_2}] = [\underbrace{[(s_0^{z_1}, a_0^{z_1}, z_1), ..., (s_T^{z_1}, a_T^{z_1}, z_1)]}_{\text{"Put a mug in the coffee machine"}}, \underbrace{[(s_0^{z_2}, a_0^{z_2}, z_2), ...]}_{\text{"Push the brew button"}}]. \quad (1)$$

This results in a dataset $\mathcal{D}^L = \{\tau_{\bar{z}_1}, \tau_{\bar{z}_2}, \ldots\}$ of diverse trajectories with natural language descriptions (see Figure 2, left).

**Approach Overview.** Our approach leverages skill descriptions in $\mathcal{D}^L$ as instructions for the policy during pre-training: each description defines a task and the policy is rewarded for successfully executing the instruction. Intuitively, the more diverse the set of language instructions during pre-training, the more semantic skills the policy will learn and the more downstream tasks can it finetune on efficiently. Thus, SPRINT introduces two approaches for increasing the diversity of the pre-training instructions without additional human inputs. Firstly, SPRINT leverages pre-trained language models to aggregate the human annotated instructions (Figure 2, middle). Secondly, we propose an approach for cross-trajectory skill-chaining, that allows us to combine behaviors across multiple training trajectories to generate completely unseen instructions (Figure 2, right). Both approaches expand the set of pre-training tasks, leading to more effective pre-training. Once the policy is trained, we can transfer it to a new task for finetuning, either by providing a language description of the target task or by replacing the language input with a vector of trainable parameters and finetuning it jointly on the target task, like prior work (Chebotar et al., 2021; Laskin et al., 2021).

## 4.1 INSTRUCTION-CONDITIONED OFFLINE RL

To pre-train our policy with the natural language instruction dataset $\mathcal{D}^L$, we take inspiration from goal-conditioned RL approaches (introduced in Section 3): instead of conditioning the policy on goal states and rewarding it for reaching these states, we condition our policy $\pi(a|s, z)$ on *language instructions* $z$ from $\mathcal{D}^L$ and provide a sparse reward to the agent for reaching the end-state $s_T$ of the

respective sub-trajectory. Formally, we define the reward as:

$$R(s, a, z) = \begin{cases} 1, & \text{for } s = s_T^z \\ 0, & \text{otherwise.} \end{cases} \quad (2)$$

We train our policy $\pi(a|s, z)$ and a critic $Q(s, a, z)$ on this reward function with offline RL. In this work, we use Implicit Q-Learning (Kostrikov et al., 2022) since we found it to be easy to tune.

## 4.2 LANGUAGE-MODEL-BASED INSTRUCTION AGGREGATION

**LLM Prompt Example**

Summarize the following steps.

1: Pick up the tomato slice.
2: Heat it up in the microwave.
Summary: Microwave a tomato slice.

1: [SKILL 1]
2: [SKILL 2]
...
Summary:

Figure 3: A shortened example of the LLM prompt. Full prompt in Section B.

Large language models (LLMs), trained on massive corpora of internet text data, have been shown to be effective at performing a variety of tasks – from question answering to program synthesis – when prompted with relevant text (Devlin et al., 2018; Brown et al., 2020; Wang & Komatsuzaki, 2021; Rae et al., 2021; Hoffmann et al., 2022; Zhang et al., 2022; Chowdhery et al., 2022). Here we use LLMs to *aggregate*, *i.e.*, paraphrase, the existing language annotations in $\mathcal{D}^L$ (see Figure 2, middle, for an illustration). Given a trajectory that contains multiple sub-trajectories, we can aggregate adjacent sub-trajectories into a longer trajectory and relabel its natural language annotation with a summary of the individual instructions generated by the LLM, thereby generating a new *higher-level* pre-training task that encompasses instructions from multiple sub-trajectories.[1] We use a simple summarization prompt to instruct the language model (see Figure 3 for an example, Section B for the full prompt). Specifically, we aggregate with OPT-13B (Zhang et al., 2022), an open-source 13 billion parameter LLM comparable with the second-largest GPT-3 text completion model, text-curie-001 (Brown et al., 2020).[2] Like before, the reward for this new aggregated sub-trajectory is 1 at the last transition and 0 otherwise. Taking Eq. 1 as an example, we prompt the LLM to summarize the two skills ($z_1$ : "*Put a mug in the coffee machine*," $z_2$ : "*Push the brew button*"), resulting in a new semantic annotation $\hat{z}_{1:2}$ describing both skills (*e.g.*, "*Make coffee*"). We then add the new trajectory back to our dataset $\mathcal{D}^L$. Using this technique, we generate new language annotations for all $\binom{N}{2}$ tuples of consecutive sub-trajectories in our dataset. In practice, this allows us to increase the number of pre-training task instructions by 2.5x without additional human effort.

## 4.3 CROSS-TRAJECTORY CHAINING

Agents trained with offline RL can combine behaviors from multiple trajectories via value propagation, *i.e.*, "stitch" them (Levine et al., 2020). For example, if trajectory (A) shows cleaning the mug in the sink and then placing it on the shelf, while trajectory (B) starts with placing the mug in the coffee machine, offline RL algorithms are able to learn to clean the mug in the sink and then place it in the coffee machine (see Figure 2, right). In our case of *instruction-conditioned* offline RL, enabling such stitching behavior requires special care. Due to the different language instruction conditionings for the critic $Q(s, a, z_A)$ and $Q(s, a, z_B)$, values do not naturally propagate from trajectory (B) back to trajectory (A). Instead, we must actively add "chaining examples" to our training dataset (Chebotar et al., 2021). For this, we randomly sample a sub-trajectory $\tau_{z_A} = [s_{0:t}, a_{0:t}, z_A]$ from the training dataset $\mathcal{D}^L$, *e.g.*, part of trajectory (A), and *replace* its language instruction with the instruction $z_B$ *from another trajectory* $\tau_{z_B}$, *e.g.*, "*place mug in coffee machine*." Note that we do not need to sample the full trajectory (A). Instead, $s_t$ can be any state from trajectory (A) from which we try to execute skill (B). Crucially, we *cannot* use the same reward function as before: since the last state $s_t$ of the sampled sub-trajectory, *e.g.*, a mug in the sink, does not solve the instruction $z_B$, "*put mug in coffee machine*," we cannot set its reward to 1. Which reward should we use instead?

Let's recap that when using temporal-difference (TD) learning (Sutton & Barto, 2018), Q functions for the sparse reward definition from Equation 2 intuitively represent a value that is proportional to

---

[1]Other relabeling operations, such as splitting an instruction into lower-level instructions, can also be performed by the LLM. However, such operations require grounding the LLM to the agent's observations to determine sub-trajectory split points. We leave investigating this to future work.

[2]See appendix, Sections E.1 and E.2 for example aggregated skills and comparisons of different LLMs.

the probability of reaching the goal at time $T$ (Eysenbach et al., 2022; Chebotar et al., 2021):

$$Q^\pi(s_t, a_t, z) = \mathbb{E}\left[\sum_{t'=t} \gamma^{t'} R(s_{t'}, a_{t'}, z)\right] = \mathbb{E}\left[\gamma^{T-t} \mathbb{1}\left[s_T = g_z\right]\right] \propto P^\pi(s_T = g_z | s_t, a_t). \quad (3)$$

Similarly, we want $Q(s, a, z_B)$ to represent the probability of reaching the goal $g_{z_B}$, the mug being in the coffee machine, from state $s$. Thus, we need to set the reward of the last state $s_t$ in the sampled sub-trajectory to the probability of reaching the goal $g_{z_B}$ from $s_t$, *i.e.*, $Q(s_t, a_t, z_B)$:

$$R(s, a, z_B) = \begin{cases} Q(s, a, z_B), & \text{for } s = s_T \\ 0, & \text{otherwise.} \end{cases} \quad (4)$$

Finally, we can apply the skill aggregation approach from Section 4.2 in the cross-trajectory case. Instead of sampling two sub-trajectories from the same training trajectory as in skill aggregation, we can now sample $\tau_{z_A}$ and $\tau_{z_B}$ from *different* training trajectories and chain them together. The aggregate instruction $\hat{z}$ implies that the agent first finishes skill (A) and then finishes skill (B), *e.g.*, "*clean the coffee mug and place it in the coffee machine*." Thus, following the logic above, we need to set the reward for the final state in trajectory (B), $s_{T_B}$, to 1 since it solves the task $\hat{z}$, and the reward in the final state of trajectory (A), $s_{T_A}$, to the probability of solving skill (B) from there:

$$R(s, a, \hat{z}) = \begin{cases} 1, & \text{for } s = s_{T_B} \\ Q(s, a, z_B), & \text{for } s = s_{T_A} \\ 0, & \text{otherwise.} \end{cases} \quad (5)$$

Note that here, unlike in Section 4.2, we do not construct a combined trajectory from $\tau_{z_A}, \tau_{z_B}$, as we do not know the states and actions required to transition from the last state of (A) to states in (B). See appendix Section C.4 for a more detailed discussion on chaining. To generate the aggregate instruction $\hat{z}$ we can use the LLM summarization from Section 4.2. Yet, in practice we found the resulting summaries are often meaningless, since randomly paired instructions *from different trajectories* can rarely be summarized meaningfully. Instead, we saw better performance by simply concatenating the natural language instructions from $\tau_A$ and $\tau_B$. Finally, since $Q$ changes over the course of training, we compute the rewards in Eqs. 4 and 5 online during training. We perform relabeling of trajectories at equal proportions for both rewards.

## 5 EXPERIMENTS

In our experiments, we investigate how well an agent pre-trained with SPRINT performs on unseen, semantically meaningful tasks. Specifically, we are interested in answering the following questions: (1) Does pre-training with SPRINT lead to more efficient finetuning on unseen target tasks than previous pre-training approaches? (2) Can agents pre-trained with SPRINT execute unseen language instructions zero-shot? (3) Does pre-training with *semantic* skills lead to better generalization to unseen environments than unsupervised pre-training approaches?

### 5.1 EXPERIMENTAL SETUP

**ALFRED Benchmark.** We choose the ALFRED benchmark (Shridhar et al., 2020) for evaluation (see Figure 4), since in contrast to existing multi-task RL environments (Yu et al., 2019), it allows us to test our pre-trained agents on *semantically meaningful*, *long-horizon* tasks in a realistic household setting. Additionally, ALFRED provides a dataset of 6.6k episodes of offline agent experience with crowd-sourced natural language instruction annotations (*e.g.*, "*pick up the dirty mug*", "*wash it in the sink*", ..., see Figure 4). The observation space in ALFRED consists of $300 \times 300$ egocentric RGB observations and the action space consists of 12 discrete action choices (e.g. *turn left*, *look up*, *pick up object*), along with 82 discrete object types (Pashevich et al., 2021). We leverage 73k natural language instructions and their associated observation-action trajectories from the dataset for pre-training. For more environment and dataset details, see Section D.

**Evaluation Tasks.** We create a set of 100 varied-horizon evaluation tasks (*EVAL_100*), by sampling sequences of 1 to 7 instructions from the ALFRED dataset and asking human annotators to provide a high-level language instruction that describes the sequence. For example, the 5-instruction sequence

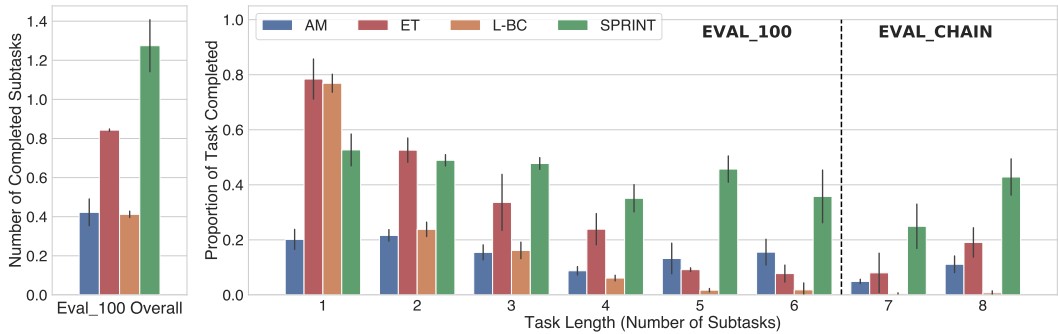

Figure 5: Zero shot evaluation performance on the *EVAL_100* and *EVAL_CHAIN* task sets. The left figure plots the overall number of completed subtasks on *EVAL_100* (*i.e.*, average return), with our method in green. The right figure plots the proportion of task completed (number of subtasks successfully executed divided by task length), split by task length, in both the task sets. See appendix, Table 3 for exact numerical results.

"pick up the knife", "slice the potato", "put the knife in the fridge", "pick up a potato slice", "put the slice in the fridge" was summarized as "Cut and refrigerate the potato on the counter" (for more examples, see appendix, Section D.2). The task is solved if an agent completes all 5 instructions, in order, conditioned on the human task description. We ensure that *EVAL_100* has an equal distribution of length 1, 2, 3, 4, and 5+ tasks (20 of each category). Additionally, we create a set of 20 evaluation commands that test the agent's long-horizon chaining capabilities (*EVAL_CHAIN*): we withhold trajectories and instruction summaries of sequences with 7 to 8 instructions from the ALFRED dataset, making sure that the respective skill chains are *not* present in any single training trajectory. Thus, the agent needs to learn to chain behaviors *across multiple trajectories* to solve these tasks. Finally, to test environment generalization capabilities, we also create a set of 10 tasks in household floor plans not seen in any trajectory in the training dataset (*EVAL_UNSEEN*), consisting in equal parts of 1 to 5-instruction sequences.

**Comparisons.** We compare SPRINT against common policy pre-training approaches: behavioral cloning and offline goal-conditioned RL. Specifically, we compare to the following prior works:

- **Language-conditioned BC (L-BC)** (Jang et al., 2021; Lynch & Sermanet, 2021): Behavior cloning (BC) conditioned on the individual ALFRED language instructions.

- **Episodic Transformers (ET)** (Pashevich et al., 2021): A transformer BC architecture that conditions on full sequences of language instructions—SOTA on certain ALFRED tasks.

- **Actionable Models (AM)** (Chebotar et al., 2021): Goal-conditioned offline RL with randomly sampled goal observations from the ALFRED training set.

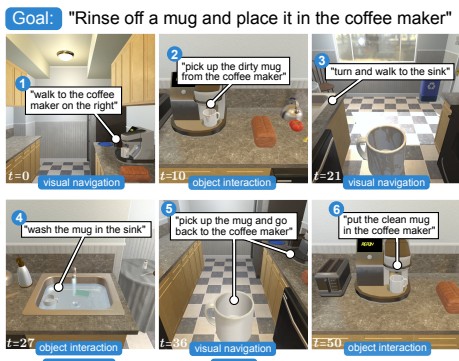

Figure 4: ALFRED. Figure drawn from Shridhar et al. (2020), with permission.

We implement all methods with the same architecture and hyperparameters where possible, and pre-train them for the same number of steps with the same language tokens.. For more implementation details, see appendix Section C. All results reported are means and standard deviations over 3 seeds.

### 5.1.1 SPRINT SOLVES LONG-HORIZON TASKS ZERO-SHOT

We first test the effectiveness of SPRINT's pre-training by analyzing zero-shot performance across the 100 tasks in the *EVAL_100* evaluation set. We report results in Figure 5 (left). Our approach, SPRINT, achieves 3x higher zero-shot task performance than prior pre-training approaches AM

and L-BC. Even though ET is designed for ALFRED evaluations and benefits from a transformer architecture, SPRINT still outperforms it by 1.5x. To better understand the differences between the methods, we report the breakdown of returns by length of the evaluation task in Figure 5, right. We find that L-BC achieves good performance on short-horizon tasks. However, on long-horizon tasks, SPRINT achieves much higher returns, since it can leverage the language-model to automatically generate longer-horizon pre-training tasks. In contrast, standard L-BC approaches train only on the human-provided, shorter-horizon annotations and thus cannot zero-shot perform long-horizon tasks. This trend holds even when evaluating with detailed step-by-step instructions (see appendix Section E.3). Similar to our approach, AM trains to reach long-horizon goals during pre-training but the results in Figure 5, right, show that AM's pre-training with goal-state conditioning is *less* effective than our language-conditioned pre-training.

To evaluate the pre-training approaches' ability to perform *unseen* skill sequences, we evaluate zero-shot performance on the *EVAL_CHAIN* task set in Figure 5. Unsurprisingly, both L-BC and AM largely fail to generate returns on these long-horizon tasks. ET does slightly better as it is trained to attend to long-horizon instruction sequences, but it still performs poorly on these tasks as it cannot generalize to the higher-level human-written task descriptions. In contrast, SPRINT can solve unseen long-horizon tasks with up to 8 subtasks (see Figure 7 for an example trajectory, appendix Section E.5 for more qualitative comparisons). This shows that, with the help of cross-trajectory chaining, SPRINT is able to go beyond the skills in the training data and execute unseen tasks.

### 5.1.2 SPRINT AGENTS FINETUNE EFFECTIVELY IN UNSEEN ENVIRONMENTS

To test downstream task performance, we finetune the pre-trained agents on the *EVAL_UNSEEN* task set in unseen household floor plans. To implement finetuning for SPRINT and AM, we condition the policy on a language instruction or goal image from the target task respectively and then run IQL with online data collection. For L-BC, we first pre-train a language-conditioned Q-function with IQL on the pre-training dataset and then finetune policy and Q-function with online IQL. We do not finetune ET as it is a transformer architecture that is nontrivial to implement for RL.

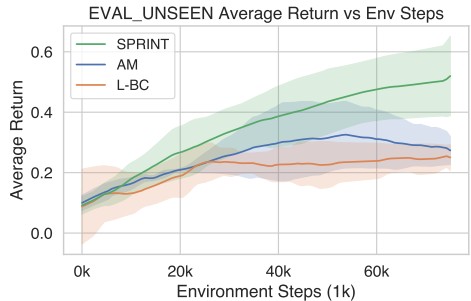

Figure 6: Finetuning results on floorplans not seen during pre-training.

We report finetuning results in Figure 6, with qualitative examples in appendix, Section E.5. Our approach, SPRINT, is able to achieve 2x higher downstream task return than the best prior work. Specifically, we find L-BC converges quickly to a low return as it is not pre-trained for longer horizon tasks. Meanwhile, AM struggles to transfer the learned skills to the new environment, possibly since the goal states from the new environment are unseen. In contrast, our method's pre-training with language conditioning allows for effective transfer even to unseen environments since the semantics of the tasks transfer well: the language description "*place cup in coffee machine*" transfers to many environments while the goal image for the same task might look very different. Thus, pre-training with semantic language instructions can enable better transfer for learning tasks in new environments than pre-training to reach goal states.

### 5.1.3 ABLATION STUDIES

We verify the effectiveness of the components of our approach, with the following ablations:

- **SPRINT w/o chain**: removes cross-trajectory chaining (Section 4.3), instead trains only on within-trajectory human-provided and LLM-aggregated tasks
- **SPRINT w/o LLM-agg**: additionally removes the LLM aggregation (Section 4.2), thus trains offline RL agent only on the human-provided task annotations.

We report zero-shot evaluation results in Table 1. The results show the importance of the different components of our approach. Without cross-trajectory chaining, performance drops by ∼20-30%. Interestingly, we find that performance also degrades on the *EVAL_100* task set, indicating that cross-trajectory chaining can improve performance even on long-horizon tasks in the training set.

Task: "Warm up a piece of apple"

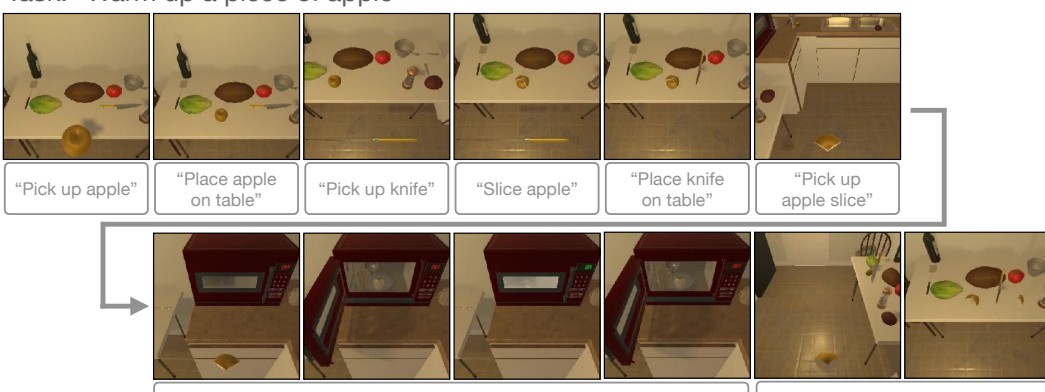

Figure 7: Example task execution of our pre-trained SPRINT agent for the task "*Warm up a piece of apple*". Successful execution of this task requires solving 8 subtasks in sequence and a total of 50 steps. This exact sequence of subtasks was never observed in the training data. SPRINT leverages cross-trajectory stitching and LLM aggregation to learn how to execute unseen tasks.

We observe an even larger performance loss when removing the language model aggregation of pre-training tasks: the resulting agent is solely trained on the shorter horizon human task annotations, and thus struggles to solve long-horizon evaluation tasks, similar to the L-BC approach in Section 5.1.1. We provide the

Table 1: *EVAL_100* and *EVAL_CHAIN* returns.

| | *EVAL_100* | *EVAL_CHAIN* |
|---|---|---|
| SPRINT (ours) | **1.27 ± 0.13** | **2.59 ± 0.66** |
| SPRINT w/o chain | 0.91 ± 0.03 | 2.04 ± 0.04 |
| SPRINT w/o LLM-agg | 0.38 ± 0.05 | 0.10 ± 0.04 |

per-task-length breakdown of the ablation performances in appendix, Section E.4. This section also presents additional ablation experiments that use simple concatenation instead of LLM summarization for task aggregation, resulting in the same longer-horizon pre-training trajectories but different task labels, further supporting the importance of the usage of LLMs in our pre-training approach.

## 6 DISCUSSION, LIMITATIONS, AND FUTURE WORK

In this paper, we presented SPRINT, an approach for scalable agent pre-training with offline instruction-conditioned RL. We demonstrated how we can leverage easy-to-collect natural language instructions on offline agent experience for effective pre-training. Further, we introduced approaches for using pre-trained large language models and cross-trajectory skill chaining to automatically expand the set of pre-training tasks. In our experimental evaluations on the ALFRED household task benchmark (Shridhar et al., 2020) we demonstrated that SPRINT pre-training allows for substantially more efficient finetuning on downstream tasks in unseen environments than prior works. SPRINT represents a step towards scalable approaches for pre-training agents with a set of semantically meaningful skills.

Naturally, pre-training with semantic skills will be most effective on long-horizon, semantically meaningful tasks. Many of the common RL benchmarks like MetaWorld (Yu et al., 2019) and D4RL (Fu et al., 2020) do not evaluate agents on semantically meaningful or long-horizon tasks and thus we do not expect our method to show significant pre-training benefits there. However, with the recent trend towards evaluating agents on more realistic, semantically meaningful tasks, like BEHAVIOR (Srivastava et al., 2022) or MineDojo (Fan et al., 2022), we believe semantic pre-training approaches like SPRINT will be beneficial. Another limitation of our current approach is that SPRINT can *only* leverage language-annotated data. While we can collect such annotations at scale via crowd-sourcing, there will always be a lot of agent experience data *without* language annotations. Investigating approaches that can *combine* language-annotated and un-annotated data for effective pre-training is an interesting direction for future work.

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

APPENDIX

# Table of Contents

## A  SPRINT PSEUDOCODE

---

**Algorithm 1** SPRINT Algorithm

---

**Require:** Dataset $\mathcal{D}^L$ w/ language instruction labels, LLM
1: AGGREGATESKILLS($\mathcal{D}^L$, LLM)       ▷ Automated LLM skill relabeling (Sec. 4.2)
2: **while** not converged **do**
3:   $\tau_z \leftarrow \mathcal{D}^L$: Sample an annotated skill (sub-)trajectory
4:   Train offline RL on $\tau_z$
5:
6:   $\tau_z^{\text{chain}} \leftarrow$ CROSSCHAINSKILLS($\mathcal{D}^L$)      ▷ Cross-trajectory skill chaining (Sec. 4.3)
7:   Train offline RL on $\tau_z^{\text{chain}}$
8:
9:   $\tau_{\text{agg}_1}, \tau_{\text{agg}_2} \leftarrow$ CROSSAGGREGATESKILLS($\mathcal{D}^L$, LLM)    ▷ Cross-traj. aggregation (Sec. 4.3)
10:   Train offline RL on $\tau_{\text{agg}_1}, \tau_{\text{agg}_2}$
11: **end while**
12:
13: **procedure** AGGREGATESKILLS($\mathcal{D}^L$, LLM)
14:   **for** composite trajectory $\tau_{\bar{z}}$ in $\mathcal{D}^L$ **do**
15:    Separate $\tau_{\bar{z}}$ into language annotated sub-trajectories $[\tau_{z_1}, ..., \tau_{z_N}]$
16:    **for** all adjacent sub-trajectories $[\tau_{z_i}...\tau_{z_j}]$ **do**
17:     Assign name from LLM: $\text{LLM}(z_i...z_j) = \hat{z}_{i:j}$
18:     $\tau_{\hat{z}_{i:j}} \leftarrow$ Concat $[\tau_{z_i}, ..., \tau_{z_j}]$ and relabel with $\hat{z}_{i:j}$
19:     $T \leftarrow$ length of $\tau_{\hat{z}_{i:j}}$
20:     $R(s_T, a_T, \hat{z}_{i:j}) = 1$       ▷ Label last reward with 1 (Eq. 2).
21:     $\mathcal{D}^L = \mathcal{D}^L \cup \{\tau_{\hat{z}_{i:j}}\}$
22:    **end for**
23:   **end for**
24: **end procedure**
25:
26: **procedure** CROSSCHAINSKILLS($\mathcal{D}^L$)
27:   Sample random (sub)trajectories $\tau_{z_1}, \tau_{z_2} \sim \mathcal{D}^L$
28:   Sample random endpoint $j$ in $\tau_{z_1}$
29:   $\tau_{z_2}^{\text{chain}} \leftarrow [(s_0, a_0, 0, z_2), ..., (s_j, a_j, Q^\pi(s_j, a_j, z_2), z_2)]$    ▷ Relabel reward w/ Eq. 4.
30:   **return** $\tau_{z_2}^{\text{chain}}$
31: **end procedure**
32:
33: **procedure** CROSSAGGREGATESKILLS($\mathcal{D}^L$, LLM)
34:   Sample random (sub)trajectories $\tau_{z_1}, \tau_{z_2} \sim \mathcal{D}^L$ with lengths $T_1, T_2$
35:   Assign new name : $\hat{z} =$ "$\{z_1\}.\{z_2\}$"
36:   $\tau_{\text{agg}_1} \leftarrow [(s_0, a_0, 0, \hat{z}), ..., (s_{T_1}, a_{T_1}, Q^\pi(s_T, a_T | z_2), \hat{z})]$   ▷ Relabel reward w/ Eq. 5
37:   $\tau_{\text{agg}_2} \leftarrow [(s_0, a_0, 0, \hat{z}), ..., (s_{T_2}, a_{T_2}, 1, \hat{z})]$     ▷ Relabel reward w/ Eq. 5.
38:   **return** $\tau_{\text{agg}_1}, \tau_{\text{agg}_2}$
39: **end procedure**

---

## B  LARGE LANGUAGE MODEL PROMPT

We list the full large language model summarization prompt in Figure 8. The examples in the prompt are fixed for all summarization queries. These examples are selected from the ALFRED validation dataset (which is not otherwise used in our work) at random: We spell out the primitive skill annotations in the "Task Steps:" part of each prompt example. Then, the "Summary" for each of these is the high-level, human-written annotation for that trajectory from ALFRED. We repeatedly sampled these trajectories until each example mentioned a different object to prevent biasing the LLM towards certain types of objects.

We note that the "Look at the vase under the light" example is important to make the LLM give reasonable summaries for similar tasks in ALFRED where the agent picks something up and turns on a light. This is because most of the human labels for turning on the lamp do not mention the object in the previous step, making it difficult for the LLM to realize that the task has to do with looking at the held object under a lamp.

Instructions: summarize the following ordered steps describing common household tasks.

Task Steps: 1: Pick up the smaller knife on the counter to the left of the stove. 2: Slice the tomato with the smaller knife. 3: Put the knife in the sink. 4: Pick up a slice of tomato from the countertop. 5: Heat up the slice of tomato in the microwave, removing it afterwards.
Summary: Microwave the tomato slice after slicing it with the smaller knife on the counter.

Task Steps: 1: Pick up the vase. 2: Turn on the lamp.
Summary: Look at the vase under the light.

Task Steps: 1: Grab the pencil off of the desk. 2: Put the pencil in the bowl. 3: Grab the container off of the desk. 4: Put the container down at the back of the desk.
Summary: Put a bowl with a pencil in it on the desk.

Task Steps: 1: Pick up the bar of soap from the back of the toilet. 2: Put the bar of soap in to the sink, turn on the faucet to rinse off the soap, pick up the soap out of the sink. 3: Put the soap in the cabinet under the sink and on the left.
Summary: Put a rinsed bar of soap in the cabinet under the sink.

Task Steps: 1: [SKILL 1]. 2: [SKILL 2]. 3: [SKILL 3]. ... N: [SKILL N].
Summary:

Figure 8: The full prompt that we use for summarization. Following the suggestions of Ahn et al. (2022) for prompt design, we explicitly number each step. The LLM completion task begins after "Summary:". For brevity, we omit the new line characters between all numbered steps.

## C  BASELINES AND IMPLEMENTATION

We implement IQL (Kostrikov et al., 2022) as the base offline RL algorithm for all goal-conditioned offline RL pretraining baselines and ablations due to its strong offline and finetuning performance on a variety of dense and sparse reward environments. At a high level, IQL trains on in-distribution $(s, a, s', r, a')$ tuples from the dataset rather than sampling a policy for $a'$ to ensure that the Q and value functions represent accurate estimated returns constrained to actions in the dataset. The value function is trained with an expectile regression loss controlled by a hyperparameter $\tau$, where $\tau = 0.5$ results in standard mean squared error loss and $\tau \to 1$ approximates the max operator, resulting in a more optimistic value function that can better "stitch" together trajectories to obtain distant reward in sparse reward settings. The IQL policy is trained to maximize the following objective:

$$e^{\beta(Q(s,a)-V(s))} \log \pi(a|s),$$

which performs advantage-weighted regression (Peng et al., 2019) with an inverse temperature term $\beta$. In practice, the exponential advantage term is limited to a maximum value to avoid numerical overflow issues. We detail shared training and implementation details below, with method-specific information and hyperparameters in the following subsections.

**Observation space.** The state space of the ALFRED environment consists of $300 \times 300$ RGB images. Following the baseline method in ALFRED (Shridhar et al., 2020), we preprocess these images by sending them through a frozen ResNet-18 encoder (He et al., 2016) pretrained on ImageNet (Deng et al., 2009). This results in a $512 \times 7 \times 7$ feature map that we use as the observation input to all networks. Furthermore, as ALFRED is a partially observable, egocentric navigation environment, we concatenate the last 5 frames as the full observation, resulting in an observation that is of the shape $(512 * 5) \times 7 \times 7$.

**Action space.** The agent chooses from 12 discrete low-level actions. There are 5 navigation actions: MoveAhead, RotateRight, RotateLeft, LookUp, and LookDown and 7 interaction actions: Pickup, Put, Open, Close, ToggleOn, ToggleOff, and Slice. For interaction actions the agent additionally selects one of 82 object types to interact with, as defined by Pashevich et al. (2021). In total, the action space consists of $5 + 7 * 82 = 579$ discrete action choices. Note that this action space definition is different from the action space in Shridhar et al. (2020), which

used a pixel-wise mask output to determine the object to interact with. In contrast to Shridhar et al. (2020) we aim to train agents with *reinforcement learning* instead of imitation learning and found the discrete action parametrization more amenable to RL training than dense mask outputs. For all methods, due to the large discrete action space, we perform some basic action masking to prevent agents from taking actions that are not possible. For example, we do not allow the agent to `Close` objects that aren't closeable nor can they `ToggleOn` objects that can't be turned on.

**Language embedding.** We embed language instructions with the `all-mpnet-base-v2` pretrained sentence embedding language model from the SentenceTransformers python package (Reimers & Gurevych, 2019). This produces a 768-dimensional language embedding which is used as input for language-conditioned policy and critic functions, as detailed below.

**Policy and critic networks.** We train a discrete policies with two output heads of size 12 and 82 for the action and interaction object outputs respectively. Critic networks are conditioned on both the observation and the discrete action output of the policy. In all policy and critic networks, we process the ResNet feature observation inputs with 4 convolutional layers. In networks with language input, we flatten the output of the convolutional layers and concatenate the observation features with the 768-dim language embedding, before passing the concatenated image-language features through a series of fully connected layers. Additionally, we use FiLM (Perez et al., 2018) to condition convolutional layers on the language embeddings.

**Pre-training hyperparameters.** A hyperparameter search was performed first on the language-conditioned BC-baseline to optimize for training accuracy. These hyperparameters were carried over to the IQL implementation, and another search for IQL-specific hyperpameters were performed on a baseline IQL policy conditioned on semantic instructions. With these parameters fixed, we performed one more hyperparameter search specific to Actionable Models but for the final implementation of SPRINT we re-used the same hyperparameters and only selected SPRINT-specific parameters heuristically. Hyperparameters for each method are detailed in a separate table. Shared hyperparameters for all methods (where applicable) are listed below:

| Param | Value |
| --- | --- |
| Batch Size | 1024 |
| # Training Batches | 140k |
| Learning Rate | 2e-3 |
| Optimizer | AdamW |
| Dropout Rate | 0.2 |
| Weight Decay | 0.05 |
| Discount $\gamma$ | 0.98 |
| Q Update Polyak Averaging Coefficient | 0.005 |
| Q-Network Discrete Action Embedding Size | 48 |
| Q-Network Discrete Object Selection Action Embedding Size | 24 |
| Policy and Q Update Period | Once every training iteration |
| Batch Norm | True |
| Nonlinearity | ReLU |
| IQL Advantage Clipping | [0, 100] |
| IQL Advantage Inverse Temperature $\beta$ | 10 |
| IQL Quantile $\tau$ | 0.9 |

**Finetuning details and hyperparameters.** We perform finetuning experiments for Language-conditioned BC, Actionable Models, and SPRINT. For all models, we finetune the model on only newly collected task data by running online IQL (without any of the chaining or aggregation steps). Each method is finetuned on every task in the *EVAL_UNSEEN* task set individually; that is, we pre-train once and then finetune ten times, once for every task in the task set. We then average returns over all tasks, then report metrics averaged over all random seeds. For each task, we define a maximum rollout time horizon of 2 timesteps per environment action required by the expert planner.

When not specified, finetuning parameters are identical to pre-training parameters. Finetuning hyperparameters are specified below:

| Param | Value |
|---|---|
| Dropout Rate | 0 |
| # Initial Rollouts | 50 |
| Training to Env Step Ratio | 0.5 |
| $\epsilon$ in $\epsilon$-greedy action sampling | 0.25: annealed down to 0.05 |
| # Parallel Rollout Samplers | 4 |

## C.1 LANGUAGE-CONDITIONED BEHAVIOR CLONING

Our language-conditioned behavior cloning (L-BC) comparison method is inspired by and replicates BC-Zero (Jang et al., 2021) and LangLfP (Lynch & Sermanet, 2021). BC-Zero performs FiLM-conditioned semantic imitation learning (Perez et al., 2018) and both BC-Zero and LangLfP have an additional image/video-language alignment objective. In BC-Zero, their video alignment objective aligns language embeddings with videos of humans performing tasks related to those the BC-Zero robot agent trains on. LangLfP's image-language alignment objective allows their policy to accept both image and natural language goals as input due to only having a subset of their data labeled with hindsight language labels. As we don't have human videos of these tasks and our entire dataset is labeled with language labels, we do not add a video or image alignment objective.

Therefore, we implement L-BC by using the same architecture as described above with just a single policy network that trains to maximize the log-likelihood of actions in the dataset. As our entire dataset consists of expert trajectories, this baseline ideally learns optimal actions for the instructions.

Hyperparameters for the L-BC baseline are identical to the shared parameters above, where applicable.

## C.2 EPISODIC TRANSFORMERS

Episodic Transformers (ET) (Pashevich et al., 2021) trains a transformer architecture on full sequences of ALFRED instructions with a behavior cloning objective. This is currently state of the art in the "Seen Path-Length Weighted Success Rate" evaluation metric on the ALFRED leaderboard. We used the ET implementation from the official code repository and used the pre-tuned hyperparameters.

For fair comparison, we make a few modifications to make it as close as possible to SPRINT and the baselines: 1) we train it on the same dataset as all baselines, so we do not generate new synthetic training data like Pashevich et al. (2021) does in the original implementation, 2) we encode visual frames with a Resnet-18 instead of Resnet-50 backbone, 3) we use a context window of 5 frames just like for other policies, 4) we remove the high-level goal specification from the input text tokens as we do not assume access to those, and 5) we train the model for longer to match the number of training steps for all methods.

## C.3 ACTIONABLE MODELS (AM)

Actionable Models (Chebotar et al., 2021) pretrains a goal-conditioned Q function conditioned on randomly sampled image goals and also performs a goal-chaining procedure very similar to our semantic skill chaining procedure. We implement AM by modifying the base IQL policy and critic networks to take in image goals instead of natural language embeddings as goals. These goals are provided in the same way as the observations, i.e., as a concatenated stack of 5 frames (the last 5 frames in the trajectory) processed by a frozen ResNet-18. Therefore, goals are the same shape as observations: $(512 * 5) \times 7 \times 7$.

To allow for fair comparison between our approach and AM, we implement AM with the same powerful offline RL algorithm, IQL (Kostrikov et al., 2022), used in our method. IQL ensures that the policy does not choose out-of-distribution actions by using advantage-weighted regression on in-distribution actions for policy extraction. With this, we found the conservative auxiliary loss AM adds to push down Q-values for out-of-distribution actions to be unnecessary and even hurtful to its overall performance, so we omit this additional loss term.

We also pre-train AM on the same long-horizon trajectories as those generated by SPRINT during LLM-based skill aggregation. This ensures a fair comparison in terms of the types and lengths of tasks seen during pre-training.

Finally, after consulting the authors of AM, we tried varying maximum trajectory lengths when sampling random goals. We found that allowing random goals to be sampled from anywhere within a trajectory resulted in the best zero-shot evaluation performance for AM, so our numbers are reported with this implementation detail.

## C.4 SPRINT

The implementation details of SPRINT follow from the discussion about implementing IQL at the top of this section. The key differences are in (1) language model skill aggregation and (2) cross-trajectory skill chaining, detailed below.

**LLM Skill Aggregation.** We perform LLM skill aggregation fully offline by iterating through every ALFRED trajectory and aggregating sequences of adjacent primitive skill sub-trajectories. Assuming a trajectory with $N$ primitive skills, we select all $\binom{N}{2}$ pairs of start and end skills and aggregate all instructions from start to end with the LLM. With 73k original language-annotated trajectories, this procedure allows us to generate an additional 110k aggregated trajectories. We then add these trajectories to the original dataset and train on the entire set.

**Cross-trajectory skill chaining implementation.** We perform cross-trajectory skill chaining in-batch. Instead of sampling a second trajectory to perform chaining on, we simply permute the batch indicies to generate a set of randomly sampled second trajectories. Then, we perform a second loss function update, in addition to the original update on the sampled trajectories, with equal loss weighting, to apply the skill-chaining update. We apply both chaining procedures from Eq. 4 and Eq. 5 in-batch with equal weight. Empirically, we found that cross-trajectory skill chaining works slightly better with the on-policy Value function obtained through IQL, therefore we use state values at the chaining targets instead of state-action Q-values.

SPRINT-specific hyperparameters follow:

| Param | Value |
|---|---|
| Large Language Model for Relabeling | OPT-13B (Zhang et al., 2022) |
| LLM Token Filtering Top-p[1] | 0.9 |
| LLM Token Sampling Temperature | 0.8 |

[1]At each token generation step, only the highest probability tokens with total probability mass that add up to the top-p are considered.

**Cross-trajectory chaining extended discussion.** When performing cross-trajectory chaining, particularly using Eq. 5, special care must be taken to preserve the dynamics of the original MDP. When chaining together two trajectories $\tau_A$ and $\tau_B$, we concatenate the two sentences of each trajectory together and relabel their rewards with Eq. 5. The new language annotation used to chain together these trajectories is the concatenation of the two sentences, implying that the agent finishes skill (A) and then skill (B). However, we cannot concatenate the two trajectories together into one longer trajectory, as doing so would imply that the agent can instantaneously jump from the last state of skill (A) to the first state of skill (B), which may not be possible. Therefore, we instead treat the relabeled trajectories at separate trajectories with the same language annotation (lines 36 and 37 of Algorithm 1).

However, this introduces two possible complications: 1) Language annotations differing in structure from those in the original dataset, and 2) Possible instruction ambiguity:

1. **Language annotations differing in structure.** Language annotations produced by the chaining procedure will result in annotations that implicitly skip certain steps. For example, when chaining skill (A), "make the bed," and skill (B), "make a cup of coffee," the resulting chained annotation will be "Make the bed. Make a cup of coffee." However to perform skill (B) the agent needs to first move to the kitchen from the bedroom to make the cup of coffee, which is skipped in this annotation. LLM-based skill aggregation (Section 4.2) helps bridge this gap by summarizing long-horizon sequences while skipping certain implied steps. For example, one real LLM summary (listed in Figure 14) summarized the sequence: "*1: Pick up the plaid pillow that is on the left end of the couch. 2: Place the pillow on the ottoman*" into the instruction "*Place a plaid pillow on the ottoman*," which skipped the step of picking up the pillow as it is implied that you must do so before placing the pillow down. Using the LLM augments our original dataset such that, in ALFRED, we have 2.5X more data after performing offline skill aggregation. Therefore after performing LLM aggregation, there are many examples of similar instructions to those used for chained trajectories that imply certain steps without mentioning them explicitly.

2. **Instruction ambiguity.** When chaining trajectories, there will be some ambiguity introduced as we do not have intermediate instructions for going from the last state of A to the initial state of B (obtaining these instructions requires additional human effort). This ambiguity is only present in the states of trajectory A, as in trajectory B we presume that the instructions for trajectory A are finished and the agent then can just follow the instructions relevant for trajectory B. We believe that the effects of the ambiguity on pre-training performance depends greatly on the given dataset. In complex and diverse environments like ALFRED, hindsight-labeled annotations will contain details specific to certain scenes, resolving this ambiguity. In ALFRED, the annotations usually contain information about the specific objects that the agent must interact with or locations that the agent must go to. For example, annotations for rinsing mugs typically are of the form "*clean the MUG in the sink,*" or annotations for picking up a candle will often say something like "*pick up the YELLOW CANDLE on the COUNTER,*" highlighting specific details regarding what the agent is supposed to do to complete the trajectory.

# D DATASET AND ENVIRONMENT DETAILS

## D.1 DATASET DETAILS

For training and evaluation we leverage the ALFRED benchmark and dataset (Shridhar et al., 2020). The ALFRED training dataset contains ∼6.6k trajectories collected by an optimal planner following a set of 7 high-level tasks with randomly sampled objects (*e.g.*, pick up an object and heat it). Each trajectory has at least three crowd-sourced sets of language instruction annotations. Each trajectory consists of a sequence of 3-19 individually annotated skills (see Figure 9, left). This results in a total of 141k language-annotated skill trajectories.

However, nearly half of the language instructions in the ALFRED dataset are navigation skill instructions like "turn left, then look up and walk to the counter on the right". To get a more balanced skill annotation dataset, we merge all navigation skills with the skill that immediately follows them, using only the annotation of the next skill. After this processing step, the resulting dataset contains 73k language-annotated primitive skill trajectories. After we merge the navigation skills, the average number of skills in each trajectory is 3.5 skills per trajectory (Figure 9, middle), and the average number of actions in each skill is 14.3 (Figure 9, right).

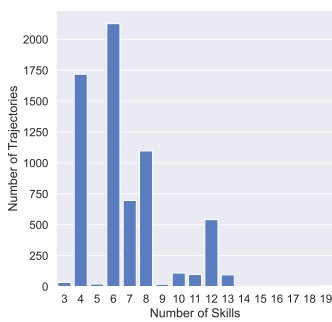 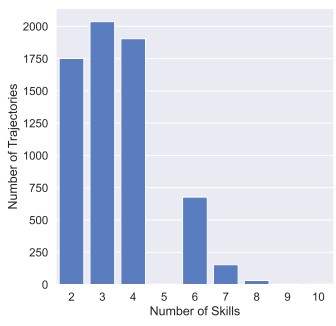 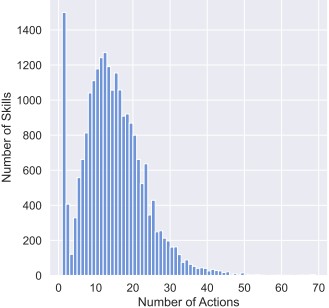

(a) Skills per trajectory in the original ALFRED dataset.

(b) Skills per trajectory in the merged dataset.

(c) Actions per skill in the merged dataset.

Figure 9: **Left**: distribution of the number of skills in each trajectory in the original ALFRED dataset. **Middle**: distribution of skills per trajectory in the "merged" dataset with merged navigation skills. **Right**: distribution of number of actions per skill in the "merged" dataset.

## D.2  EVALUATION TASKS

**Data Collection Overview**

Thank you for participating in this short summarization task. You will be writing one-sentence summaries of ordered instructions describing household tasks. Please try to ensure that the one-sentence summary you write either implicitly or explicitly describes the entire set of instructions.

For example, given the following sentences:

```
1. Open the fridge and take a pot of water out of the fridge, then close the fridge door.
2. Boil the pot of water on the stove.
```

You could write "Boil a pot of water from the fridge."

A few more examples:

```
1. Cut the lettuce to the left of the sink.
2. Put the knife down on the stove.
3. Take a slice of lettuce.
Summary: Cut the lettuce on the left of the sink and take a slice.

1. Pick up the glass of water to the right of the sponge, on the left of the shelf.
2. Pour the glass of water into the plant's soil.
3. Fill the glass with water from the sink.
4. Pour the glass of water into the plant.
5. Put down the glass of water.
Summary: Water the plant with two glasses of water.
```

You will be writing at most 40 summaries for instruction sets ranging from 2 to up to 5 instructions long.

During this process, if you feel like a given set of instrutions can't be readily turned into a one-sentence high-level summary, feel free to get a new set of sentences by pressing the "Skip" button. Once you understand these instructions, feel free to continue to the next cell and begin!

**Annotation Task 1 (40 summaries)**

```python
In [1]: from sam_train_valid_data_collection_utils import interact_program
        train_annotated_skills = []
        scene_type="train"
        interact_program(train_annotated_skills, scene_type)
```

Skills to summarize:  Please write a one-sentence task description that describes the following instructions:

1. Pick up the bottle on the toilet basin.
2. Place the bottle behind the bar soap on the counter.

Write your summary here:

Skip

Submit Summary

Figure 10: Data collection jupyter notebook page. Note that there is a "Skip" button so that human annotators can skip an instruction sequence if they do not feel it is semantically meaningful or easy to summarize.

We evaluate agents through zero-shot policy evaluation and finetuning on three sets of evaluation tasks in the ALFRED environment: (1) *EVAL_100* to measure the ability of pre-trained agents to execute semantically meaningful instructions at varied levels of abstraction, (2) *EVAL_CHAIN* to measure the ability of agents to chain behaviors across multiple trajectories to solve long tasks, and (3) *EVAL_UNSEEN* to evaluate generalization performance when finetuning to unseen household floor plans.

**Collecting evaluation task data.**  The ALFRED dataset provides high-level language annotations for each of the trajectories in the dataset. We could use these annotations as unseen task-instructions to evaluate our agents. However, we found that the different skills are not equally distributed across trajectories of different skill lengths, *e.g.*, most 2-skill trajectories perform pick-and-place tasks while tasks involving heating skills only appear in length 7+ trajectories. To allow evaluation with a less biased skill distribution, we create the *EVAL_100* task set by randomly choosing a trajectory from the ALFRED dataset *and then randomly sampling a subsequence of skills of a certain length* from this trajectory. To obtain a high-level language instruction that summarizes this new subsequence, we crowd-source labels from human annotators. For labeling, each annotator is presented

with a remotely hosted Jupyter notebook interface (see Figure 10). Whenever we by chance sample a full ALFRED trajectory for annotation, we directly used the existing high-level annotation from the ALFRED dataset. We annotate 80 trajectories with human annotators and combine them with 20 randomly sampled single-skill trajectories, resulting in a total of 100 evaluation tasks (see Figure 11 for example instructions). This results in 20 tasks of length 1 skills, 20 tasks of length 2 skills, 20 tasks of length 3 skills, 20 tasks of length 4 skills, and 20 tasks of lengths 5+ (5-7) skills.

For *EVAL_CHAIN*, we randomly sampled 20 full trajectories from the ALFRED dataset that had sequences of 7 or 8 skills (10 of length 7, 10 of length 8) and removed these trajectories from the post-LLM aggregated training dataset. We did not remove any of the LLM-aggregated trajectories made up of subsequences of skills within that trajectory. This allows AM and SPRINT to perform skill chaining to solve these tasks by ensuring that there were valid sequences of skills to chain together to be able to solve these removed tasks. For example, assume a (shortened for clarity) sampled skill sequence is "pick up apple," then "put apple in microwave", then "slice the apple." Then, either Actionable Models or SPRINT can chain together the sub-trajectory associated with "pick up apple" then "put apple in microwave" with the "slice the apple" sub-trajectory to solve this task. These trajectories all had annotations from ALFRED annotators, so we used those annotations directly (see Figure 13 for example instructions).

Finally, for *EVAL_UNSEEN*, we collected a set of 10 full-length trajectories from the ALFRED "valid-unseen" dataset consisting of validation tasks in unseen floor plans. We collected 2 of each length from 1 through 5 for a total of 10 tasks by sampling random full-length trajectories from this dataset, with the exception of length 1 tasks (we just sample random skills to create length 1 tasks). As these are full trajectories, they already have human annotations from ALFRED, which we directly use as the task description (see Figure 12 for example instructions).

We list additional details about the tasks in each evaluation set in Table 2.

Table 2: Evaluation Task Specifics. Note that the "number of env actions per task" corresponds to the number of environment actions the ALFRED expert planner required to complete that task.

|  | *EVAL_100* | *EVAL_CHAIN* | *EVAL_Unseen* |
|---|---|---|---|
| Number of Tasks | 100 | 20 | 10 |
| Task Lengths (# primitive skills) | [1, 2, 3, 4, 5, 6, 7] | [7, 8] | [1, 2, 3, 4, 5] |
| Min Number of Env Actions per Task | 1 | 34 | 2 |
| Avg Number of Env Actions per Task | 39.1 | 60.9 | 46.6 |
| Max Number of Env Actions per Task | 113 | 104 | 124 |

Finally, we display 5 randomly sampled tasks, along with their human annotations, from each of our task sets in Figures 11, 12, and 13.

**Online finetuning environment setup.** During online-finetuning we initialize the agent in the same house floor plan as the trajectory the task was extracted from to ensure executability. During finetuning, we give each episode a time horizon of 2x the number of environment actions needed by the expert planner to solve the task. We give sparse rewards for each skill solved by the agent during the episode. Therefore for length 1 tasks, the agent can only be rewarded once before the episode ends, while for length 5 tasks, the episode terminates on the fifth reward signal. We give a reward of $\frac{1}{\text{num total skills}}$ for each skill the agent successfully executes so that the return sums to 1. We found that this helped to finetune all comparison methods more stably, possibly due to the fact that giving larger rewards (*e.g.*, 1 for each skill) results in out-of-distribution critic values (when compared to pre-training) that de-stabilize online reinforcement learning.

Skills to Summarize: 1: Grab the knife on the counter. 2: Place the knife in the sink then turn the faucet on so water fills the sink. Turn the faucet off and pick up the knife again. 3: Place the knife on the table to the left of the wooden bowl.
Annotator Summary: Wash the knife from the counter, put in on the table.

Skills to Summarize: 1: Pick up the blue book closest to your and the phone from the bed. 2: Turn on the lamp to take a look at the book in the light.
Annotator Summary: Examine the book by the light of a lamp.

Skills to Summarize: 1: Pick up yellow candle on counter. 2: Open cabinet, put candle in cabinet, close cabinet 3: Pick up yellow candle from toilet.
Annotator Summary: Move the candle from the sink to the cabinet under the sink, close it and and then pick the candle from the top of the toilet in front of you.

Skills to Summarize: 1: Pick the pot on the left side up from the stove. 2: Set the bowl and knife on the table next to the tomato.
Annotator Summary: Put the bowl with the knife in it next to the tomato.

Skills to Summarize: 1: Pick up the pen that's in front of you that's under the mug. 2: Put the pencil in the mug that was above it. 3: Pick up the mug with the pencil in it.
Annotator Summary: Put the pen into the mug and pick up the mug.

Figure 11: Randomly sampled, human language instruction annotations from the *EVAL_100* task set.

Skills to Summarize: 1: Pick up the lettuce on the counter. 2: Chill the lettuce in the fridge. 3: Put the chilled lettuce on the counter, in front of the bread.
Annotator Summary: Put chilled lettuce on the counter.

Skills to Summarize: 1: Pick up an egg from off of the kitchen counter. 2: Open the fridge, put the egg in to chill for a few seconds and then take it back out. 3: Place the cold egg in the sink.
Annotator Summary: Chill an egg and put it in the sink.

Skills to Summarize: 1: Pick up the butter knife off of the right side of the kitchen island. 2: Put the knife handle down in the frying pan that is on the front left burner of the stove. 3: Pick up the frying pan with the knife in it off of the stove. 4: Put the frying pan with the knife in it into the sink basin to the right of the potato.
Annotator Summary: Put a frying pan with a knife in it into the sink.

Skills to Summarize: 1: Take the pencil from the desk. 2: Put the pencil on the desk.
Annotator Summary: Take the pencil from the desk, put it on the other side of the desk.

Skills to Summarize: 1: Pick up the left pillow on the chair. 2: Put the pillow on the sofa right of the newspaper. 3: Pick up the pillow on the chair. 4: Put the pillow on the sofa left of the newspaper.
Annotator Summary: Place two pillows on a sofa.

Figure 12: Randomly sampled, human language instruction annotations from the *EVAL_UNSEEN* task set.

Skills to Summarize: 1: Pick up the knife in front of the lettuce. 2: Slice the apple in the sink with the knife. 3: Place the knife into the sink. 4: Pick up the sliced apple from the sink. 5: Place the apple slice into the pot on the stove. 6: Pick up the pot from the stove. 7: Pick up the pot from the stove.
Annotator Summary: Slice an apple for the pot on the stove and put the pot on the counter to the right of the door.

Skills to Summarize: 1: Take the apple from the counter in front of you. 2: Place the apple in the sink in front of you. 3: Take the knife by the sink in front of you. 4: Cut the apple in the sink in front of you. 5: Place the knife in the sink in front of you. 6: Take an apple slice from the sink in front of you. 7: Heat the apple in the microwave, take it out and close the microwave. 8: Place the apple slice in the sink in front of you.
Annotator Summary: Place a warm apple slice in the sink.

Skills to Summarize: 1: Pick up the loaf of bread. 2: Put the bread on the counter above the spatula. 3: Pick up the knife that's above and to the right of the loaf of bread. 4: Cut the top half of the loaf of bread into slices. 5: Put the knife on the edge of the counter in front of you horizontally. 6: Pick up a slice of bread from the middle of the loaf. 7: Cook the bread in the microwave then take it out and close the microwave door. 8: Throw the cooked slice of bread away.
Annotator Summary: Put a microwaved slice of bread in the oven.

Skills to Summarize: 1: Pick the knife up from off of the table. 2: Open the microwave, slice the potato, and close the microwave. 3: Open the microwave, place the knife inside of it, and close the microwave. 4: Open the microwave, pick up the potato slice inside, close the microwave. 5: Place the potato slice in the pan on the stove. 6: Pick up the pan from the stove. 7: Open the refrigerator, place the pan inside, and close the refrigerator.
Annotator Summary: Move the pan from the stove top to inside the black refrigerator.

Skills to Summarize: 1: Pick up the red tomato on the counter to the right of the stove. 2: Put the tomato onto the island below the butter knife. 3: Pick up the butter knife off of the kitchen island. 4: Slice up the tomato on the kitchen island. 5: Place the butter knife onto the island to the right of the sliced tomato. 6: Pick up a tomato slice off of the kitchen island. 7: Open the fridge and put the tomato slice on the bottom shelf, then close the door, after a couple seconds open the fridge and remove the tomato slice then close the door. 8: Open the microwave door and place the tomato slice inside the microwave in front of the egg.
Annotator Summary: Put a chilled tomato slice into the microwave.

Figure 13: Randomly sampled, human language instruction annotations from the *EVAL_CHAIN* task set.

# E  EXTENDED EXPERIMENTS, RESULTS, AND ANALYSIS

Table 3: *EVAL_100* and *EVAL_Chain* eval dataset per-length and overall skill completion rates. See Section 5 for experiment setup.

|  |  | AM | L-BC | SPRINT | ET |
|---|---|---|---|---|---|
| *EVAL_100* | Number of Completed Subtasks Overall | $0.46 \pm 0.05$ | $0.41 \pm 0.02$ | $\mathbf{1.27 \pm 0.13}$ | $0.84 \pm 0.01$ |
|  | Length 1 Progress | $0.23 \pm 0.01$ | $0.77 \pm 0.04$ | $0.53 \pm 0.07$ | $\mathbf{0.78 \pm 0.09}$ |
|  | Length 2 Progress | $0.22 \pm 0.03$ | $0.24 \pm 0.03$ | $0.49 \pm 0.03$ | $\mathbf{0.53 \pm 0.06}$ |
|  | Length 3 Progress | $0.17 \pm 0.03$ | $0.16 \pm 0.04$ | $\mathbf{0.48 \pm 0.03}$ | $0.33 \pm 0.12$ |
|  | Length 4 Progress | $0.10 \pm 0.01$ | $0.06 \pm 0.01$ | $\mathbf{0.35 \pm 0.06}$ | $0.24 \pm 0.07$ |
|  | Length 5 Progress | $0.16 \pm 0.08$ | $0.02 \pm 0.01$ | $\mathbf{0.46 \pm 0.06}$ | $0.09 \pm 0.01$ |
|  | Length 6 Progress | $0.16 \pm 0.01$ | $0.02 \pm 0.03$ | $\mathbf{0.36 \pm 0.12}$ | $0.08 \pm 0.04$ |
|  | Length 7 Progress | $0.00 \pm 0.00$ | $0.00 \pm 0.01$ | $\mathbf{0.01 \pm 0.02}$ | $0.00 \pm 0.00$ |
| *EVAL_CHAIN* | Number of Completed Subtasks Overall | $0.67 \pm 0.09$ | $0.04 \pm 0.05$ | $\mathbf{2.59 \pm 0.66}$ | $1.04 \pm 0.35$ |
|  | Length 7 Progress | $0.04 \pm 0.01$ | $0.00 \pm 0.00$ | $\mathbf{0.25 \pm 0.10}$ | $0.09 \pm 0.00$ |
|  | Length 8 Progress | $0.13 \pm 0.02$ | $0.01 \pm 0.01$ | $\mathbf{0.43 \pm 0.08}$ | $0.03 \pm 0.01$ |

Here, we present additional results complementary to the experiments in the main paper in Section 5. We present and analyze LLM annotation examples in Section E.1, zero-shot evaluations with step-by-step task instructions in Section E.3, and an extended ablation analysis in Section E.4.

## E.1  LLM SUMMARY EXAMPLES

We randomly sample 12 OPT-13B task summaries produced while performing skill aggregation (explained in Section 4.2) using the prompt in Figure 8 and display them in Figure 14. Because we sample the tokens sequentially during text generation, some of the summaries mostly describe one or a few of the skills but in a non-sensical manner (*e.g.*, LLM(*"Pick up the plastic bowl on the left. Turn on the lamp."*) = *"Look at the plastic bowl on the left"*). The summaries also likely can be improved by including more examples in the prompt. However, after analyzing many more examples, we see that OPT-13B generally provides fitting high-level summaries for most sequences by skipping over implied sub-tasks (although it sometimes also skips over important sub-tasks, likely due to the prompt). For example, it summarizes the task *"Pick up the plaid pillow that is on the left end of the couch. Place the pillow on the ottoman"* as *"Place a plaid pillow on the ottoman,"* in which picking up the plaid pillow is implied. It also often demonstrates object relation understanding, *e.g.*, by stating *"Place a plaid pillow"* when only the first sub-task in the sequence mentions that the pillow is plaid.

Skills to Summarize: 1: Pick up the plaid pillow that is on the left end of the couch. 2: Place the pillow on the ottoman.
LLM-generated Summary: Place a plaid pillow on the ottoman.

Skills to Summarize: 1: Pick up the piece of potato furthest to the right. 2: Open the microwave, place the potato inside, close the door, microwave for two seconds, then remove the potato. 3: Place the cooked potato piece to the right of the salt shaker.
LLM-generated Summary: Place a cooked potato piece on the table.

Skills to Summarize: 1: Put the spoon in the pot on the left. 2: Pick up the pot with the spoon.
LLM-generated Summary: Put the pot with the spoon on the left.

Skills to Summarize: 1: Open the microwave above the stove and slice the apple inside with the knife. 2: Place the knife standing up on the blue dish. 3: Remove a slice of apple from the microwave. 4: Place the apple slice on the blue plate.
LLM-generated Summary: Put an apple slice on the blue plate.

Skills to Summarize: 1: Pick up the butter knife that is next to the fork. 2: Open the refrigerator. 3: Open the refrigerator and place the knife on the bottom shelf in the refrigerator then close the door.
LLM-generated Summary: Open the refrigerator and place the butter knife on the bottom shelf in the refrigerator then close the door.

Skills to Summarize: 1: Open the fridge and put the knife inside. 2: Take a piece of the apple out of the microwave.
LLM-generated Summary: Take a slice of apple out of the microwave.

Skills to Summarize: 1: Pick up the remote that is on the shelf. 2: Put the remote on the couch.
LLM-generated Summary: Put the remote on the couch.

Skills to Summarize: 1: Pick up the dirty green towel to the right of the bar of soap on the vanity. 2: Put the towel into the sink, rinse it for a few seconds and pick it back up. 3: Open the far left bottom drawer to the left of the sink, put the rinsed green towel inside and close the drawer.
LLM-generated Summary: Put the towel in the drawer.

Skills to Summarize: 1: Pick up the plastic bowl on the left. 2: Turn on the lamp.
LLM-generated Summary: Look at the plastic bowl on the left.

Skills to Summarize: 1: Slice up the fruit that is in the sink. 2: Place the knife in the second row of the fridge. 3: Grab a slice of the fruit. 4: Place the fruit slice on the back right pan. 5: Grab the pan that has the fruit in it.
LLM-generated Summary: Put a slice of fruit on a pan in the fridge

Skills to Summarize: 1: Pick up the pot on the front right burner. 2: Chill the pot in the fridge then get it back out and close the door.
LLM-generated Summary: Put a pot on the stove, put it in the fridge, get it out, close the door.

Skills to Summarize: 1: Pick up the knife on the counter. 2: Cut the potato in the sink into slices. 3: Put the knife in the sink with the potato. 4. Pick up a slice of potato from in the sink. 5. Warm the potato slice in the microwave and then remove it.
LLM-generated Summary: Warm the potato slice in the microwave.

Figure 14: LLM Summary examples. We randomly sample 12 language combinations generated by the OPT-13B text generation model (Zhang et al., 2022) and display them without cherry-picking specific examples. The LLM often creates a high level description by skipping implied steps (like humans do), *e.g.*, "Put the remote on the couch" implies the need to pick up the remote.

### E.2    COMPARING SUMMARIES FROM DIFFERENT LLMS

In this section, we justify the use of OPT-13b as our summarization model by comparing its summaries of three different skill sequences to OPT-350m (350 million parameters) and OPT-1.3b (1.3 billion parameters). The summaries and sequences are displayed in Figure 15. Although these are 3 randomly sampled examples, they highlight differences between the models that hold true across many examples that we examined.

As the language model size increases, it understands more of the relationships between sub-tasks in each sequence, therefore generating better summaries. An example of this is the second example in Figure 15, which highlights a sequence in which both OPT-1.3b and OPT-13b understand the high-level task of putting a warmed plate in the cabinet, as opposed to OPT-350m which essentially repeats the second sub-task in the sequence. Yet sometimes, only OPT-13b produces a reasonable summary. In the last example, OPT-13b produces a summary that implies that an agent must wash the potato ("Put a *rinsed* potato on the counter..."); the other two models' summaries only mention the last step of putting the potato down.

---

Skills to summarize: 1. Pick the clock up from the desk. 2. Turn the lamp on.
**OPT-350m Summary:** Look at the clock.
**OPT-1.3b Summary:** Turn on the lamp.
**OPT-13b Summary:** Look at the clock.

Skills to summarize: 1. Take the plate from the cabinet to the left of the stove. 2. Warm the plate in the microwave, above the stove. 3. Put the plate in the cabinet.
**OPT-350m Summary:** Place the plate in the microwave.
**OPT-1.3b Summary:** Put a warmed plate in the cabinet.
**OPT-13b Summary:** Put a warmed plate in the cabinet.

Skills to summarize: 1. Open the microwave, take out the potato and shut the door. 2. Put the potato in the sink, turn the water on and off again. Take the potato out of the sink. 3. Place the potato on the counter, to the right of the sink.
**OPT-350m Summary:** Put the potato on the counter, to the left of the sink.
**OPT-1.3b Summary:** Place the potato on the counter, to the right of the sink.
**OPT-13b Summary:** Put a rinsed potato on the counter to the right of the sink.

---

Figure 15: Comparison of OPT-350m, 1.3b, and 13b summaries on 3 randomly sampled sequences from our dataset. In general, as the model size increases, the summary becomes better. However, there are some sequences all three on which all 3 models do not generate good summaries for, such as the first sequence in these examples.

### E.3    STEP BY STEP ZERO-SHOT EVALUATIONS

Humans may sometimes prefer to give more detailed instructions, *e.g.*, if they do not trust a household robot to successfully execute a high-level instruction. In that case, they are likely to give detailed, step-by-step instructions instead. In this section, we demonstrate the effectiveness of our pre-training strategy in enabling execution of longer-horizon tasks when specified by such step-by-step instructions. To generate these step-by-step instructions, we combine all subtask instructions into one paragraph to condition agents on instead of the high-level human annotation. For example, a task with 2 subtasks, "Pick up the knife" and "Slice the potato," is given the simple task annotation "Pick up the knife. Slice the potato."

We compare SPRINT and IL on both *EVAL_100* and *EVAL_CHAIN* tasks with the combined, step-by-step task annotations in Table 4. L-BC and SPRINT both perform similarly on the step-by-step *EVAL_100* tasks compared to with the original annotations, while both perform better on

*EVAL_CHAIN* tasks thatn with the original annotations. This is likely due to how *EVAL_CHAIN* tasks are on average much longer (7 or 8 tasks). Therefore, instructions spelling out exactly what to do allow agents to more easily understand what and in which order subtasks should be performed.

Table 4: Step-by-step evaluation number of completed sub-tasks (number of completed sub-tasks).

|  | L-BC | SPRINT |
|---|---|---|
| *EVAL_100* Completed Subtasks | $0.46 \pm 0.01$ | $\mathbf{1.24 \pm 0.19}$ |
| *EVAL_CHAIN* Completed Subtasks | $0.22 \pm 0.03$ | $\mathbf{3.11 \pm 0.08}$ |

### E.4 EXTENDED ABLATION STUDY

In this section, we examine a series of additional questions regarding specific design choices of our method, in a manner complementary to Section 5.1.3. We also visualize all ablations' zero-shot policy evaluation performance in on both *EVAL_100* and *EVAL_CHAIN* task sets in Table 5.

**How much does the LLM contribute to skill aggregation?** To answer this question, we compare using SPRINT with LLM aggregation on adjacent sub-trajectory sequences but no chaining (**SPRINT w/o chain**) to SPRINT with skill aggregation, but where the skills are relabeled by naïvely concatenating the sentences together (**SPRINT w/o chain, w/ concat-agg**). Across both task sets, SPRINT w/o chain, w/ concat-agg is outperformed by SPRINT w/o chain, especially in *EVAL_CHAIN*. This signifies that using the LLM helps with understanding very-long horizon, high-level semantic instructions as the LLM generates relevant task summaries for consecutive skills in a trajectory.

**What if we relabel with the LLM during chaining?** We examine also using the LLM to label skills during cross-trajectory skill aggregation (**SPRINT w/ LLM-chain**), rather than concatenating the skill annotations together (**SPRINT**). Overall, SPRINT w/ LLM-chain performs slightly worse in average return and success rates. When analyzing the summaries generated by the LLM, we found that randomly paired instructions can rarely be summarized meaningfully, thereby resulting in noisy and sometimes meaningless instructions. Therefore, we implemented SPRINT w/ LLM-chain by only utilizing the top 1% (top 10 candidates with our batch size of 1024) of in-batch chaining candidates (ranked by the LLM's prediction of what the next skill should be). Even so, the performance is not better than just simply concatenating the instructions, as even these top 1% candidates still have a high chance of not being sensible sentences to summarize.

**Does LLM skill aggregation help L-BC?** We perform LLM skill aggregation with the L-BC baseline in order to examine whether LLM skill aggregation alone can help with pure imitation learning methods (**L-BC w/ LLM-agg**). This change along doubles the average number of subtasks completed against regular L-BC in *EVAL_100* and causes it to jump from 0.04 subtasks completed in *EVAL_CHAIN* to 1.86 (Table 2 for original L-BC results). These results demonstrate the significant impact LLM aggregation alone can make.

Table 5: Zero-shot Ablation Subtasks Completion Table

|  | *EVAL_100* Subtasks Completed | *EVAL_CHAIN* Subtasks Completed |
|---|---|---|
| SPRINT (ours) | $\mathbf{1.27 \pm 0.13}$ | $\mathbf{2.59 \pm 0.66}$ |
| SPRINT w/ LLM-chain | $1.15 \pm 0.01$ | $2.46 \pm 0.21$ |
| SPRINT w/o chain | $0.91 \pm 0.03$ | $2.04 \pm 0.04$ |
| SPRINT w/o chain, w/ concat-agg | $0.77 \pm 0.06$ | $0.67 \pm 0.20$ |
| L-BC w/ LLM-agg | $0.93 \pm 0.04$ | $1.86 \pm 0.12$ |
| SPRINT w/o LLM-agg | $0.38 \pm 0.05$ | $0.10 \pm 0.04$ |

### E.5 QUALITATIVE COMPARISON RESULTS

**Zero-shot evaluation.** We compare SPRINT, AM, and L-BC zero-shot evaluation results on long *EVAL_CHAIN* tasks in Figure 16. In general, SPRINT is able to make substantially more progress

on *EVAL_CHAIN* tasks as it leverages the large language model to generate longer-horizon, semantically meaningful pre-training tasks and performs cross-trajectory chaining to learn to chain its existing dataset tasks. In the visualized examples, SPRINT is able to understand and successfully execute many of the sub-tasks implied but not directly stated by the natural language task instruction. L-BC makes very little progress on these tasks, not even understanding what the first sub-task to complete should be as the task annotation is out of distribution from what it saw while training. Finally, AM is able to make some progress on some of these tasks due to its long-horizon goal pre-training objective. However, this is less effective than our language-conditioned pre-training in such zero-shot evaluations.

**Finetuning.**    We finetune SPRINT, AM, and L-BC on *EVAL_UNSEEN* tasks, in household floorplans that were never seen while training, and visualize qualitative policy rollout examples *after finetuning* in Figure 17. In general, SPRINT is able to finetune to longer-horizon tasks while AM and L-BC both struggle with making progress on longer-horizon tasks despite receiving rewards for every completed sub-task. SPRINT's ability to complete more sub-tasks on many of the longer-horizon tasks is demonstrated in Figure 17a, while a case in which both SPRINT and AM make partial progress throughout finetuning is demonstrated in Figure 17b. We believe that AM has more trouble finetuning on these tasks than SPRINT because the task specification for AM (goal images) is out of distribution; pre-training on *semantic* tasks with SPRINT allows agents to more easily learn longer-horizon behaviors as the task specifications may still be in-distribution of the pre-training tasks that LLM skill-aggregation and skill chaining produce.

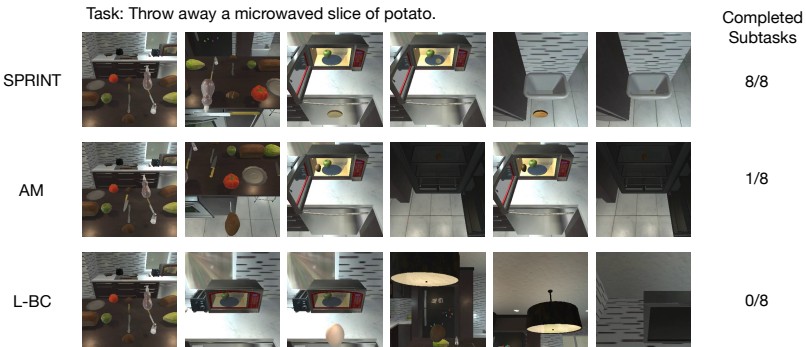

(a) SPRINT successfully solves this task, while AM fails to slice the potato and repetitively iterates between putting the potato in the fridge and microwave. L-BC fails to even pick up the potato, as the task annotation does not directly describe picking up a potato.

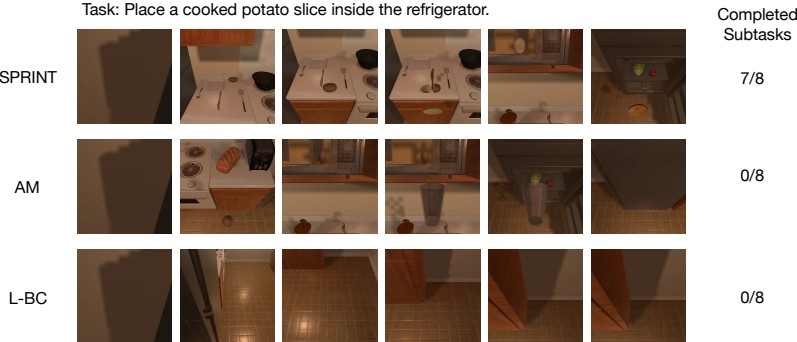

(b) SPRINT nearly solves this task, while AM picks up an egg instead of a potato. L-BC picks up random objects not related to the annotation.

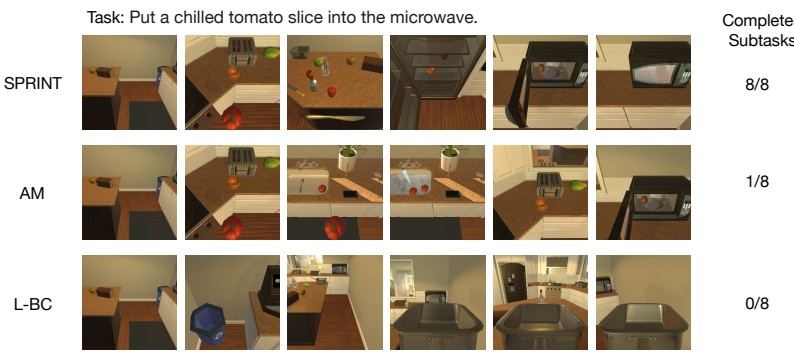

(c) SPRINT completes the entire task. AM picks up the tomato but fails to put it down onto the counter and slice it. L-BC aimlessly wanders and picks up random objects.

Figure 16: Visualizations of zero-shot policy rollouts on three tasks in the *EVAL_CHAIN* task set.

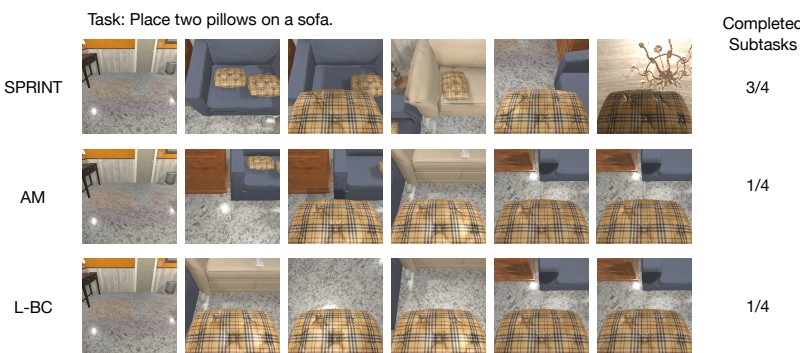

(a) SPRINT picks up and places one of the pillows on the sofa, and picks up the second but does not manage to place the second on the sofa, thus completing 3/4 subtasks. AM and L-BC both learn to pick up a pillow but never learned to place it in the correct spot.

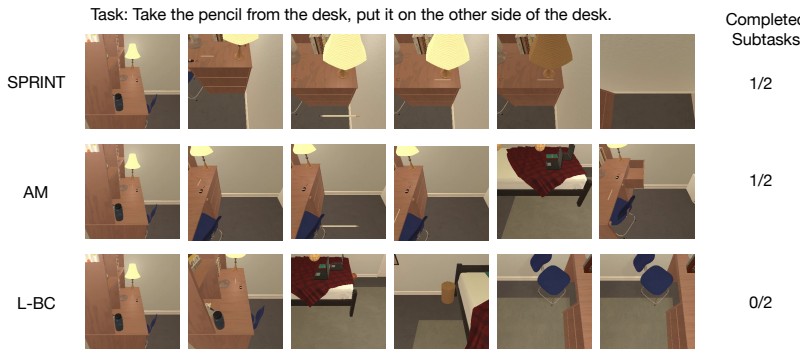

(b) SPRINT and AM both learn to pick up a pencil from the desk, although neither manage to put the pencil down in the correct place "on the other side of the desk." Meanwhile, L-BC never picks up the pencil.

Figure 17: Visualizations of policy rollouts on two tasks in the *EVAL_UNSEEN* task set, after finetuning each method. These floor plans were originally unseen to all agents until finetuning.

