# OpenReview forum: "SPRINT: Scalable Semantic Policy Pre-training via Language Instruction Relabeling"
_ICLR.cc/2023/Conference — Submitted to ICLR 2023_

### Official Review · Reviewer_QEPp · 2022-10-15

**Confidence:** 2
**Correctness:** 3
**Technical Novelty And Significance:** 2
**Empirical Novelty And Significance:** 3
**Recommendation:** 5

**Clarity, Quality, Novelty And Reproducibility:**

Clarity:

This paper is clear to read.

Quality:

This paper may have some significant concerns.

Novelty:

This paper contains novelty.
Please see the last section.

Reproductivity:

The paper does not mention (anonymous) source codes.

**Strength And Weaknesses:**

***Strength***

- It can populate pre-training data without additional human labeling. It only requires labeling sub-trajectories (maybe with crowd-sourcing), and the approach will combine the trajectories for new data.

- The evaluation results show its advantage over prior methods.

***Weakness***

**The weakness is mainly in cross-trajectory chaining.**

**1. The primary concern is the change of MDP**

The cross-trajectories chaining changes the original Markov Decision Process in the transition probability distribution P. More precisely, the value of chaining transition s_TA x a_TA x s_0B is changed.
For example, if sub-trajectory A ends in a conference room, and sub-trajectory B starts in a kitchen, then the last action of A warps the agent in the new MDP.
It encourages the agent to try warp when it wants to go to another room (after finishing a sub-trajectory).
It seems the reward of Q(s, a, z_B) is not designed to correct the problem of MDP change.

It might need explanations if the change of MDP is supposed not to have significant influences in general.

**2. The online change of Q**

As mentioned in the paper, the reward Q(s, a, z_B) changes during training and is updated online.
It requires the RL algorithm to accept online changes in rewards.
So, the paper may need more arguments on what RL algorithms are compatible (it says "any offline RL algorithm" in Section 4.1).

**3. The novelty**

- The instruction aggregation and using LLM for it might be novel. However, existing LLMs are used.

- The use of consecutive sub-trajectories seems straightforward.

- Cross-trajectory chaining has novelty, though many of its techniques are inspired by goal-conditioned RL approaches (e.g., offline goal chaining [Chebotar et al., 2021]), as mentioned in the paper.

**Summary Of The Paper:**

This paper proposes a scalable offline policy pre-training approach based on natural language instructions.

It enables automatic augmentation of pre-training data with large language models to relabel and chain across trajectories.

It empirically shows that the proposed approach outperforms prior pre-training approaches.

**Summary Of The Review:**

This paper proposes a scalable offline policy pre-training approach that outperforms conventional ones.
It combines sub-trajectories to augment data without extra human annotation.

However, it still has significant concerns, as mentioned earlier.
More explanations might help.

---

> ### Author Response · Authors · 2022-11-13
> **Response to QEPp**
>
> Thanks for your insightful review! Based on your comments, we updated our paper to better explain the cross-trajectory chaining and what RL algorithms are compatible with our approach. We answer each of your points in detail below.
>
> **“Change of MDP due to cross-trajectory chaining?”**
>
> We thank the reviewer for this insightful concern, however, the MDP is not actually changed through our objective. When performing cross-trajectory chaining, we **do not** explicitly link sub-trajectories into a longer trajectory: the agent never sees the transition $(s_{T_A}, a_{T_A}, s_{0_B})$. Instead, the final state-action-reward tuple of the relabeled sub-trajectory A is $(s_{T_A}, a_{T_A}, Q(s_{T_A}, a_{T_A}, z_B))$ and is considered a **terminal state**, therefore keeping the MDP dynamics intact. See lines 36/37 of Algorithm 1 in the appendix for exactly how we relabel these sub-trajectories.
>
> Continuing the reviewer’s example, this intuitively corresponds to finishing the skill described by sub-trajectory A in the conference room and then finishing the skill described by sub-trajectory B in the kitchen. We do not create an explicit transition between conference room and kitchen. If there is evidence in the dataset that going from the conference room to the kitchen is possible, the Q-value should be high. Otherwise, the Q-value should be low, signaling to the agent that successful execution of this skill chain is unlikely.
>
> **“What RL algorithms can handle online reward changes?”**
>
> Any actor-critic offline RL algorithm can be used with our method: in any actor-critic algorithm, the critic networks are chasing moving targets during training (computed with an updating “target network”), thus they are likely robust to rewards computed with an updating q-network too. We find this to be the case empirically, and it is further supported by prior work [Chebotar et al., 2021] that uses a similar changing reward structure, but optimizes it with a CQL-style offline actor-critic RL algorithm instead of our IQL choice, suggesting that different actor-critic offline RL algorithms are viable alternatives for optimization. We have removed mention that of “any” offline RL algorithm being compatible in Section 4.1.
>
> **“Novelty is limited”**
>
> As pointed out by the reviewer, our paper builds on prior work in goal-conditioned RL (e.g. Chebotar et al., 2021, see Section 2), yet it makes multiple crucial contributions: (1) by using language instead of goal states/images to specify tasks, our approach equips pre-trained agents with *semantically meaningful* skills. (2) Because we train with semantic skills instead of goal states / images, we can meaningfully aggregate them into new skills using large language models, substantially expanding the set of pre-training tasks. Our experiments show that these improvements lead to substantially improved 0-shot and finetuning performance over Chebotar et al (see Figures 5 and 6).
>
> Thanks again for your insightful comments! We hope that we addressed all of your points. Let us know if further clarifications are needed and please otherwise consider increasing your score.

---

> > ### Comment · Reviewer_QEPp · 2022-11-13
> > **Response**
> >
> > Thanks for the detailed explanations.
> > I have comments and extended questions.
> >
> > ***Comments***
> > The main text might make readers misunderstand that the cross-trajectory chaining approach has the same operation on sub-trajectories as the LLM-based skill relabeling approach, i.e., "aggregate adjacent sub-trajectories into a longer trajectory."
> >
> > In Figure 1, the two approaches look similar in treating sub-trajectories (also in Figure 2).
> > Also, the main text says, "we can apply the skill aggregation approach from Section 4.2 in the cross-trajectory case."
> >
> > ***Extended questions***
> >
> > The instruction for sub-trajectory B may be based on the initial state of B.
> > However, it is used as being based on the last state of A in the cross-trajectory case.
> >
> > To transit from the last state of A to the initial state of B, there should be an additional sub-trajectory. For example, "move to the kitchen" (continuing with the above example).
> >
> > So a complete instruction and a generated instruction can be different.
> >
> > A complete instruction: "*wait for 10 mins, move to the kitchen, and clean room.*"
> >
> > A generated instruction: "*wait for 10 mins and clean room.*"
> >
> > It may cause some problems.
> >
> > **1. Inconsistency with the original dataset setting.**
> > In the original setting, each sub-trajectory has an instruction. However, instructions for transitioning sub-trajectories (e.g., "move to the kitchen") are skipped in the generated instructions. So it is different from the original dataset setting.
> >
> > **2. Ambiguation.**
> > Moreover, the instruction might mean different skills in different initial states. In the example, an agent might confuse about whether it should go to the kitchen or stay in the conference room to clean room.
> >
> > It might need explanations on whether the problems are likely to happen and, if so, whether they may have significant influences.

---

> > > ### Author Response · Authors · 2022-11-14
> > > **Second response**
> > >
> > > Thank you for your prompt response! We address your comments and questions below.
> > >
> > > **“Cross-trajectory chaining similar in main text and figures”**
> > >
> > > We agree that cross-trajectory chaining could be misinterpreted based on the figures and text. Based on your comments, we have updated both Figures 1 and 2 to separate chained trajectories visually into two trajectories. We have also updated Section 4.3 with the following text: “Note that here, unlike in Section 4.2, we do not construct a combined trajectory from $\tau_{z_A}, \tau_{z_B}$, as we do not know the states and actions required to transition from the last state of (A) to states in (B).”
> > >
> > > **“Additional sub-trajectory between two chained sub-trajectories?”**
> > >
> > > When chaining trajectories, it is true that there will be some ambiguity introduced as we do not have intermediate instructions for going from the last state of A to the initial state of B (obtaining these instructions requires additional human effort). This ambiguity is only present in the states of trajectory A, as in trajectory B we presume that the instructions for trajectory A are finished and the agent then can just follow the instructions relevant to trajectory B. We believe that the effects of ambiguity on pre-training performance depend greatly on the given dataset. In complex and diverse environments like ALFRED, hindsight-labeled annotations will contain details specific to certain scenes, resolving this ambiguity. In ALFRED, the annotations usually contain information about the specific objects that the agent must interact with or locations that the agent must go to. For example, annotations for rinsing mugs typically are of the form “clean the MUG in the sink,” or annotations for picking up a candle will often say something like “pick up the YELLOW CANDLE on the COUNTER,” highlighting specific details regarding what the agent is supposed to do to complete the trajectory. We thank you for bringing up this point and have added an additional discussion regarding chaining ambiguity in Section C.4 in the appendix, and a reference to this section in the last paragraph of Section 4.3.
> > >
> > > Regarding inconsistency with the original data, as you mentioned, there is a difference—with respect to skipping implied instructions—between the original dataset language annotations and the annotations obtained through the chaining procedure. LLM-based skill aggregation helps bridge this gap by summarizing long-horizon sequences while skipping certain implied steps. For example, one real LLM summary (listed in Figure 14) summarized the sequence: “1: Pick up the plaid pillow that is on the left end of the couch. 2: Place the pillow on the ottoman” into the instruction “Place a plaid pillow on the ottoman,” which skipped the step of picking up the pillow as it is implied that you must do so before placing the pillow down. Using the LLM augments our original dataset such that, in ALFRED, we have 2.5X more data after performing offline skill aggregation. Therefore after performing LLM aggregation, there are many examples of similar instructions to those used for chained trajectories that imply certain steps without mentioning them explicitly. We have also added an additional discussion about this in Section C.4 in the appendix.
> > >
> > > -------
> > > We sincerely appreciate your detailed comments and suggestions; we believe that they have helped us greatly improve the clarity of our paper and present a more detailed discussion of the cross-trajectory chaining procedure. We hope we have addressed your concerns, and please let us know if you have any more questions. Thank you!

---

> ### Author Response · Authors · 2022-11-18
> **Response reminder**
>
> Hi, this is a gentle reminder that we have responded to your most recent reply to our review response. Thanks to your latest comments, we have:
>
> 1. Updated both main figures and the text to clarify cross-trajectory chaining better,
> 2. Answered your extended questions regarding possible label inconsistency and ambiguity from cross-trajectory chaining, and added text in the appendix talking about this in detail, along with a pointer to this section in the main text.
>
> Thanks again for your earlier response and detailed questions and comments, it has really helped us better clarify cross-trajectory chaining! We hope we have addressed all of your concerns, and if so, please consider raising your score.

---

> > ### Author Response · Authors · 2022-12-12
> > **End of discussion period response reminder**
> >
> > Hi, we just wanted to remind the reviewer that we are nearing the end of the discussion period. We appreciate your insightful comments and are grateful that they led to clarifications in our paper and an additional section in the text discussing your points in detail regarding cross-trajectory chaining. Please let us know if we have addressed all of your concerns, thank you!

---

### Official Review · Reviewer_pfEp · 2022-10-19

**Confidence:** 3
**Correctness:** 3
**Technical Novelty And Significance:** 3
**Empirical Novelty And Significance:** 3
**Recommendation:** 6

**Clarity, Quality, Novelty And Reproducibility:**

The method part is clear and easy to follow, but I have some clarification questions about experiment setups (see above).

The method seems fairly novel and interesting to me, though I cannot confidently judge the novelty due to my lack of knowledge in related work.

**Strength And Weaknesses:**

**Strength**: The proposed approach is simple and effective on authors’ test sets of ALFRED, with potential to apply to more instruction following tasks with long-horizon challenges.

**Weakness**: My main concern is that experiments are only done in one benchmark with test sets specifically created by authors that would intuitively benefit from such compositional data augmentations. Is it also possible to also provide results on standard ALFRED evaluation/test sets, which should be more than 100 selected task instances?


Some other suggestions/questions:
- Maybe mention the domain (ALFRED) and learning method (offline RL, instead of more vague “policy pre-training”) in abstract?
- Can benchmark setups be better explained? The first paragraph of Experiment sections talks about unseen task/language/environment, but then EVAL100 is said to be taken from the training set (why not eval set?), and EVALCHAIN unclear. A better explanation of task splits in original ALFRED and author created test sets is needed.
- Learning setups: Is comparison to baselines fair - do they have similar architecture (size), training steps, number of training language tokens? Are two data augmentation subschemes mixed 1:1 or other proportions? I tried to check some appendix but they are not clear, and maybe including them in main paper would be nice.
- The cross-trajectory scheme seems only applicable to offline RL, while the same-trajectory one also applies to imitation learning. Would be interesting how the latter alone might benefit IL, if authors want to claim usefulness beyond offline RL but for general policy pre-training.

**Summary Of The Paper:**

The paper proposes two data augmentation schemes for instruction following training via offline RL:

(1) **same-trajectory instruction aggregation**: Given a trajectory with multiple sub-trajectories, aggregate adjacent sub-trajectories into a longer trajectory and relabel its natural language annotation with a summary of the individual instructions. This summary is generated via a large language model (LLM) prompted with few-shot examples.

(2) **cross-trajectory chaining**: concatenate two random trajectories, and their instructions via “and”.

The proposed approach, SPRINT (Scalable Pre-training via Relabeling Language INsTructions), is evaluated on ALFRED benchmark, where authors create a set of 100 unseen
long-horizon evaluation instructions (EVAL 100), and a set of 20 evaluation commands that test the agent’s chaining capabilities (EVAL CHAIN). On these sets, SPRINT is shown to outperform imitation learning (L-BC) and offline RL (AM) baselines in zero-shot and finetuning setups.

**Summary Of The Review:**

The approach is simple, interesting, and potentially useful for more instruction following tasks, but more evaluations and explanations of setups would help better justify it.

==

After rebuttal, authors addressed many clarification questions and I raise my score from 5 to 6 in light of these.

---

> ### Author Response · Authors · 2022-11-13
> **Response to pfEp**
>
> Thank you for your thorough review! Based on your comments, we have updated our paper to clarify how we compose our evaluation task sets and included a comparison to a method that runs imitation learning on our augmented training task set. We address all your points in detail below.
>
> **“Better explain benchmark task selection”**
>
> We construct our evaluation task sets by randomly sampling instruction sequences from the ALFRED dataset and asking human annotators to summarize them in their own words, thus testing our agents on **unseen** instruction-scene combinations – we have clarified this in the text. We construct two datasets for 0-shot evaluation that differ in the tested instruction lengths: the EVAL_100 task set contains tasks with 1-7 subtasks, while EVAL_CHAIN contains tasks with 7-8 subtasks, which require agents to generalize to new instructions by **chaining** multiple skill sequences from the training data. Finally, we test finetuning on tasks in unseen scene arrangements (EVAL_UNSEEN), testing the agents’ capabilities to quickly adapt to new environments.
>
> **“Evaluate on tasks from the ALFRED test set?”**
>
> The tasks we evaluate on are representative of the tasks in the ALFRED test set: they test the agent on **unseen** instruction-scene combinations, their length and compositional structure is comparable. Like the ALFRED test set we evaluate on long-horizon tasks that require sequential execution of multiple subtasks. We cannot directly evaluate on tasks from the ALFRED test set since we require a task demonstration to compute how many subtasks the agent solved, which we don’t have for the test set tasks.
>
> **“Is comparison to baselines fair? Comparable model size, training iterations, …?**
>
> We implement all baselines with the same model architecture as our approach (i.e. same number of parameters) and train for the same number of iterations. All methods have access to the same training data, including the same language tokens for training (see Section 5.1, paragraph 4). Note that one of our contributions is to use a large language model for instruction aggregation, which augments the language annotations in the initial training dataset. We updated the last paragraph of Section 5.1 to clarify that our experiments ensure fair comparison to all baselines.
>
> **“How are the data augmentation schemes mixed?”**
>
> We perform the skill aggregation augmentation before starting training – it increases the dataset size by ~2.5X (Section 4.2, last sentence). We apply the chaining augmentation online during training for every batch, mixing within and cross-trajectory chaining at a ratio of 1:1 (we have now added this detail to Section 4.3, last sentence).
>
> **“Use skill aggregation augmentation for imitation learning?”**
>
> Thank you for this suggestion! We have added results in appendix Section E.4 for running behavioral cloning on a dataset that uses our proposed skill aggregation mechanism via large language models. We find that this indeed substantially improves the capabilities of BC for 0-shot performance of unseen tasks, verifying the effectiveness of our proposed augmentation scheme. Note however, that the performance is still worse than our offline RL approach that can also leverage our second contribution, cross-trajectory chaining. Additionally, the imitation learning approach is not amenable to fine-tuning on a downstream task and is thus not an effective pre-training approach.
>
> **“Mention domain and learning method in abstract.”**
>
> Thanks! We have incorporated your suggestion and updated our abstract accordingly!
>
> Thanks again for your insightful comments! We hope we were able to address your concerns. Let us know if any further clarifications are needed and please otherwise consider raising your score.

---

> ### Author Response · Authors · 2022-11-18
> **Response reminder**
>
> Hi, this is a gentle reminder that we have posted a detailed response to your review, added extra experiments, and updated the text according to your suggestions. Notably, we have:
>
> 1. Clarified evaluation task selection and updated the text accordingly,
> 2. Provided further details about the baselines to highlight that the comparisons are fair and updated the main text to highlight this,
> 3. Clarified the data augmentation scheme ratio in the main text,
> 4. Performed imitation learning on LLM-skill aggregated labels, per your suggestion,
> 5. Updated the abstract as you suggested.
>
> Thank you again for your detailed comments! If you have any further questions, please let us know. Otherwise, please consider raising your score.

---

### Official Review · Reviewer_bhG4 · 2022-10-25

**Confidence:** 4
**Correctness:** 4
**Technical Novelty And Significance:** 2
**Empirical Novelty And Significance:** 3
**Recommendation:** 6

**Clarity, Quality, Novelty And Reproducibility:**

The writing and figures are very clear, and the paper is of high quality. The proposed method is of mediocre novelty but the experiments are complete and well-conducted to show the advantages of proposed method.

**Strength And Weaknesses:**

> Strength

1. Generally well writen, structure of the article is well organized. Illustrative figures are intuitive and do a good job to help reader understand the main ideas. Important details are included in the appendix. Every claims of the authors are well-supported by their experiments
2. Experiments are well conducted on ALFRED household task benchmark. The results support the claimed points of the proposed. Comparisons are provided to show the superior performance of proposed methods. Ablation study is conducted to show all parts of the proposed method are useful.
3. This work has good motivation. It pretrains generalization agent with interpretable skills. The proposed method leverages easy-to-collect natural language instructions to generate unseen tasks by taking advantage of language semantics. The learned agents have skills that are semantically meaningful.

> Weaknesses

1. The idea of generating unseen language-based tasks is not novel. Previous work has already used the idea of generating new language-based tasks. The paper IMAGINE used Construction Grammar Heuristic for this job while this paper uses large language model. Therefore, the technical innovation is limited.

**Summary Of The Paper:**

The proposed method is a offline language-based goal-conditioned reinforcement learning. It needs human annotator for labeling basic tasks with languages. It implements large languages models for relabeling and cross-trajectory chaining, in order to increase the diversity of the task set. Experiments are conducted to show the proposed method exceeds previous approaches in terms of task completion rate, more efficient finetuning and 0-shot generalization.

**Summary Of The Review:**

It is a well-written paper with high completeness. The method leverages language models for efficient pretraining.

---

> ### Author Response · Authors · 2022-11-13
> **Review response to bhG4**
>
> Thank you for your insightful review!
>
> **"Previous work IMAGINE also uses language for task proposals”**
>
> We were not aware of the IMAGINE paper before, but agree that it is very relevant! Like our paper, IMAGINE [Colas et al., 2020] leverages language to formulate tasks for agent pre-training with RL. There are two key technical improvements of our method: (1) our agent is able to train via offline RL on pre-collected data, which is more sample efficient and scalable than the online RL used by IMAGINE, (2) IMAGINE requires definition of a domain-specific grammar for generating task descriptions while we leverage the capabilities of large language models to quickly adapt to a domain merely using a few example prompts, thereby reducing the required domain-specific expertise for task generation. We have included this discussion in Section 2 in our paper.
>
> Thanks again for your review and please let us know if you have any further questions!

---

> > ### Comment · Reviewer_bhG4 · 2022-12-01
> > **Thank you.**
> >
> > I'm still concerned about the novelty. I will keep the score.

---

> ### Author Response · Authors · 2022-11-18
> **Response reminder**
>
> Hi, this is a gentle reminder that we have posted a response to your review and updated our paper based on your feedback. We have updated the paper by adding your paper suggestion, IMAGINE, to the related works section and discussing differences in detail. Thanks again for your constructive review! We hope we have addressed all your concerns, and if so, please consider raising your score.

---

### Official Review · Reviewer_QcsA · 2022-10-25

**Confidence:** 3
**Correctness:** 2
**Technical Novelty And Significance:** 2
**Empirical Novelty And Significance:** 2
**Recommendation:** 5

**Clarity, Quality, Novelty And Reproducibility:**

One of the biggest concerns I had about this paper is the evaluation setting. The authors refer to their methods as a ‘pre-training method’ and claim to be able to learn new tasks more efficiently by fine-tuning on them. I found this perspective quite unconventional. Taking the ALFRED setting as an example, one trains a goal conditioned policy on training demonstrations and expects it to generalize to new tasks, which is a supervised learning problem. The authors need to motivate the setting better and make it more meaningful.

Why do the authors need to resort to offline RL approaches for this problem setting? Have you considered learning from the augmented demonstrations with imitation learning? If the proposed method is more advantageous, was the imitation learning (on the augmented data) baseline considered?

The design choices in eq(4) and eq(5) need better motivation. What is the motivation for assigning non-zero rewards to only the final state of each sub-trajectory?

The evaluation section needs better clarity for readers to understand the exact train/test setup.
The construction of EVAL_100 is vaguely described as ‘sampling sequences of 1 to 7 instructions’. I did not understand what this means (are they tasks? subgoals?).

Do the baselines benefit from your data augmentation strategies? Or are they only trained on the original set of demonstrations? I would assume the comparison is unfair in the latter case.

I found it unsatisfactory that the proposed method was not compared against state of the art methods on the ALFRED task, which makes it hard to contextualize this work. How well does the proposed method perform compared to these methods?

Figure 1 can be improved to make the approach more intuitive.


**Strength And Weaknesses:**

Pros
* Paper is generally easy to follow
* Interesting use of language models for instruction re-labeling and summarization

Cons
* Limited technical novelty
* The motivations for the paper are not clear
* Details about experimental setup are vague/missing and the results do not look particularly convincing


**Summary Of The Paper:**

This paper proposes a data-augmentation strategy to learn from demonstrations in an embodied agent setting. Two types of augmentation strategies are proposed - relabeling language instructions with a language model and changing together different tasks/trajectories to obtain a new task/trajectory. The proposed method claims to be able to learn a diverse set of skills with these augmentation strategies. The method outperforms two baselines on an evaluation set constructed by the authors based on the ALFRED dataset.

**Summary Of The Review:**

Raising score to 5 post-rebuttal.

The paper suffers from lack of strong motivations for the problem setting considered and several aspects of the experimental results are lacking.

---

> ### Author Response · Authors · 2022-11-13
> **Review response to QcsA**
>
> Thank you for your helpful comments! We have added comparisons to SOTA ALFRED imitation approaches and clarified the motivation of our method. We address each of your points below.
>
> **“Better motivate pre-training setup. Why evaluate on ALFRED?”**
>
> Pre-training RL agents for more efficient downstream learning is a widely studied problem in the RL community [Chebotar’21, Pertsch’20, Ajay’20] (see second paragraph of Section 2 for a comprehensive discussion). These works, like ours, aim to improve sample efficiency of RL agents on downstream tasks. We chose ALFRED for our evaluations since it allows us to compare different pre-training approaches on a rich set of **semantically meaningful**, **long-horizon** tasks. This is a more realistic setting than the environments used in prior pre-training RL works, which do not test long-horizon tasks [e.g. MetaWorld, Yu et al. 2020] and do not provide meaningful task sequences [e.g. FrankaKitchen, Gupta et al. 2019]. We have clarified our motivation for choosing ALFRED for our evaluations in Section 5.1.
>
> **“Why RL, not imitation? Compare to SotA ALFRED model.”**
>
> We agree with the reviewer that typically, ALFRED is used for zero-shot evaluation of imitation learning agents and to our knowledge our work is the first to perform RL in the ALFRED environment. To put our results in context, we follow your suggestion and add a comparison to a state-of-the-art imitation learning approach on ALFRED: Episodic Transformer (ET) [Pashevich et al., 2021] is #1 on the current leaderboard for success rate weighted by path length. We train ET with the same dataset and inputs as our approach (see details in Section C.2). The results in Figure 5 show that ET is superior to our simple BC baseline, but our pre-training + RL approach outperforms it. This shows the promise of RL approaches in ALFRED – enabling RL in ALFRED is an additional contribution of our work and we will release our training infrastructure and model code to encourage more (offline) RL research in ALFRED.
>
> **“Consider comparing to imitation learning on augmented dataset?”**
>
> As suggested, we add a comparison to an ablation of our approach that performs **behavioral cloning** instead of offline RL on the augmented dataset (Section E.4 in the appendix). We find that this method performs better than regular BC on zero-shot evaluation, since it is trained on a wider range of tasks in the augmented dataset. Yet, it is unable to effectively finetune, since no Q-function was learned, which demonstrates the importance of offline RL over imitation learning for enabling finetuning in our approach. Also note that even for the imitation learning ablation we still need to run our full offline RL method to generate the augmented dataset (since we require the learned Q-function for chaining).
>
> **“Why only assign reward to the final state in trajectory during chaining (eq. 4, 5)?”**
>
> We only assign rewards to the final state $s_T$ in each trajectory to accurately reflect the expected future rewards when trying to reach some goal $g$ from $s_T$. If we also assigned Q-value-based rewards to other states in the trajectory, we would effectively count the contribution of rewards from $s_T$ to g multiple times, since, e.g. the Q-value for the state $s_{T-1}$ is the sum of rewards we expect to obtain in $s_T$ + the future rewards between $s_T$ and $g$ **again**. Prior works in goal-conditioned RL have similarly chained trajectories by assigning Q-value-based rewards only to the final state of a trajectory [Chebotar et al. 2021].
>
> **“What are instructions vs tasks? How is the EVAL_100 task set constructed?”**
>
> In ALFRED, a “task” describes a sequence of language instructions (e.g. “open microwave”, “place bread inside”, “close microwave”, “heat bread”) that must be completed sequentially. We construct our evaluation task set by randomly sampling sequences of instructions from ALFRED trajectories and asking human annotators to summarize those sequences in their own words, thus creating **unseen** instructions for evaluation that test an agent’s generalization capabilities. To ensure a representative distribution of task lengths in our evaluation task set, we ensure that we evenly sample tasks with 1-7 sub-task instructions (Section 5.1, paragraph 2). We have clarified this in Section 5.1.
>
> **“Do the baselines also use the proposed data augmentations?”**
>
> No, augmenting the set of pre-training tasks via LLM-based skill relabeling and chaining is a contribution of our work (see contribution (2) in the introduction). To evaluate its effectiveness we compare to prior work that does not use such augmentations and show that our proposed augmentations lead to more effective pre-training.
>
> Thanks again for your helpful review! We hope that this addresses the concerns from your review. Let us know if further clarifications are needed and please consider raising your score otherwise.

---

> > ### Comment · Reviewer_QcsA · 2022-11-22
> > **Thank you for the response**
> >
> > I appreciate the response and the revisions.
> >
> > I do not find the argument about 'pre-training' convincing. As the authors describe, pre-training is performed in prior work for efficient downstream learning. But it is unclear to me how this work fits into that paradigm. The agents are tested on unseen instructions, but this is more of an out-of-distribution generalization problem rather than generalizing to a completely different task. As such, the positioning of this work is not clear to me.
> >
> > I appreciate the authors trying a SOTA ALFRED method on their setup. However, I would also recommend trying the proposed method on the ALFRED benchmark to better contextualize the work.
> >
> > I appreciate the additional clarifications about the method and the imitation learning baseline. Based on these additions, I am raising my score.

---

> > > ### Author Response · Authors · 2022-11-24
> > > **Response to reviewer QcsA response**
> > >
> > > Hi,
> > >
> > > Thank you for taking the time to respond to our rebuttal! We appreciate your comments and would like to clarify a few remaining points.
> > >
> > > **“Pre-training setting not convincing”**
> > >
> > > The term “pre-training” is used in different contexts across different fields of machine learning. While in computer vision or NLP it often refers to a scenario in which a network is pre-trained on one task (e.g. image classification, next token prediction) and then fine-tuned on a *completely different* task (e.g. depth prediction, sentiment analysis), policy pre-training in RL typically refers to pre-training a policy on a range of tasks in an environment and then fine-tuning it on a new (set of) tasks in variations of that environment (e.g. [1-8]). Our work similarly pre-trains on a range of ALFRED tasks and then fine-tunes on unseen tasks in unseen floor plans of the ALFRED environment. Thus we believe that, *in the RL context*, it is correct to call our approach a policy pre-training approach. The low performance of all approaches at the beginning of fine-tuning indicates that the fine-tuning tasks are substantially different from the training tasks and require major policy adjustments (Figure 6).
> > >
> > > **“Try the proposed method on the ALFRED benchmark”**
> > >
> > > Unfortunately, we cannot directly evaluate on tasks from the ALFRED benchmark test set since we require a task demonstration to compute how many subtasks the agent solved, which we don’t have for the test set tasks (also see our response to reviewer pfEp about the same question). However, the tasks we evaluate on are designed to be representative of the tasks in the ALFRED test set: they test the agent on **unseen** instruction-scene combinations and their length and compositional structure are comparable. Like the ALFRED test set, our evaluation consists of long-horizon tasks that require sequential execution of multiple subtasks. Thus, we believe that our evaluations are representative of agent performance on the ALFRED benchmark test set, although we agree with you that direct evaluation on the tasks from the test set would be ideal.
> > >
> > > Thanks again for taking the time to go through our responses – please let us know if we can provide any further clarifications!
> > >
> > > [1] Pre-Training for Robots, Kumar et al. 2022
> > >
> > > [2] Accelerating RL with Learned Skill Priors, Pertsch et al. 2020
> > >
> > > [3] Offline Primitive Discovery for Accelerating Offline RL, Ajay et al. 2020
> > >
> > > [4] Never Stop Learning: The Effectiveness of Fine-Tuning in Robotic RL, Julian et al. 2020
> > >
> > > [5] Actionable Models: Unsupervised Offline RL of Robotic Skills, Chebotar et al. 2021
> > >
> > > [6] Leveraging Broad Offline Data for Learning Visuomotor Tasks, Fang et al. 2020
> > >
> > > [7] Scalable Deep RL for Vision-Based Robotic Manipulation, Kalashnikov et al. 2018
> > >
> > > [8] Diversity is all you need (DIAYN), Eysenbach et al. 2018

---

> > > > ### Author Response · Authors · 2022-12-12
> > > > **End of discussion period response reminder**
> > > >
> > > > Hi,
> > > >
> > > > This is just a reminder that we are nearing the end of the discussion period. We appreciate your insightful comments and are excited about the changes to the paper that we have made in response. Please let us know if we can make any further clarifications about our pre-training motivation or about our evaluation procedure. If you otherwise feel that these points have been addressed, please consider raising your score.
> > > >
> > > > Thank you!

---

> ### Author Response · Authors · 2022-11-18
> **Response reminder**
>
> Hi, this is a gentle reminder that we have posted a response to your concerns and made numerous revisions to our paper based on your feedback. Namely, we have:
>
> 1. Responded to your questions regarding the motivation of pre-training then finetuning,
> 2. Added a comparison against a SOTA ALFRED method as you suggested,
> 3. Performed imitation learning on our augmented labels, as suggested,
> 4. Answered clarifying questions about the evaluation set and updated the paper accordingly.
>
> Thanks again for your constructive feedback, if you have any further questions or concerns please let us know! Otherwise, please consider raising your score.

---

### Author Response · Authors · 2022-11-13
**General Response**

We thank all reviewers for their constructive feedback! We were happy that the reviewers pointed out our paper is well motivated (bhG4), interesting (QcsA, pfEp), easy to read (QcsA, bhG4, QEPp) and has comprehensive experimental evaluations (bhG4).

We have addressed each of the reviewers’ concerns individually. Following their suggestions, we have made the following major changes to our paper:

- We include a **comparison to a SotA imitation learning approach** on the ALFRED benchmark and show that our approach outperforms it in 0-shot performance while allowing for additional finetuning
- We include a **new ablation** of our method that trains an imitation learning agent instead of an offline RL agent on our augmented dataset
- We have **clarified** how we construct the evaluation task sets for our experiments in Section 5.1 to ensure we evaluate on unseen instructions and test our agent’s generalization capabilities

All changes in the text are highlighted in orange in the updated paper draft.

---

### Decision · Program_Chairs · 2023-01-20

**Decision:**

Reject

**Justification For Why Not Higher Score:**

Not good enough.

**Justification For Why Not Lower Score:**

NA

**Metareview: Summary, Strengths And Weaknesses:**

The proposed data-augmentation strategy with LMs to learn from demonstrations in an offline RL setup is interesting has been explore in some recent studies. The reviewers expressed concerns about limited novelty, comparison with right baselines like imitation learning methods, unclear motivation of the method. Some of these concerns remain even after the discussion.




**Summary Of Ac-Reviewer Meeting:**

It was not possible to arrange a virtual meeting.